# Class-Discriminative Attention Maps for Vision Transformers

**Lennart Brocki**  *l.brocki@uw.edu.pl*
**Jakub Binda**  *j.binda@student.uw.edu.pl*
**Neo Christopher Chung**  *n.chung@uw.edu.pl*
*Institute of Informatics, University of Warsaw*

**Reviewed on OpenReview:** *https://openreview.net/forum?id=MH7xfUWHfP*

## Abstract

Importance estimators are explainability methods that quantify feature importance for deep neural networks (DNN). In vision transformers (ViT), the self-attention mechanism naturally leads to attention maps, which are sometimes interpreted as importance scores that indicate which input features ViT models are focusing on. However, attention maps do not account for signals from downstream tasks. To generate explanations that are sensitive to downstream tasks, we have developed class-discriminative attention maps (CDAM), a gradient-based extension that estimates feature importance with respect to a known class or a latent concept. CDAM scales attention scores by how relevant the corresponding tokens are for the predictions of a classifier head. In addition to targeting the supervised classifier, CDAM can explain an arbitrary concept shared by selected samples by measuring similarity in the latent space of ViT. Additionally, we introduce Smooth CDAM and Integrated CDAM, which average a series of CDAMs with slightly altered tokens. Our quantitative benchmarks include correctness, compactness, and class sensitivity, in comparison to 7 other importance estimators. Vanilla, Smooth, and Integrated CDAM excel across all three benchmarks. In particular, our results suggest that existing importance estimators may not provide sufficient class-sensitivity. We demonstrate the utility of CDAM in medical images by training and explaining malignancy and biomarker prediction models based on lung Computed Tomography (CT) scans. Overall, CDAM is shown to be highly class-discriminative and semantically relevant, while providing compact explanations.
Code available: **https://github.com/lenbrocki/CDAM**

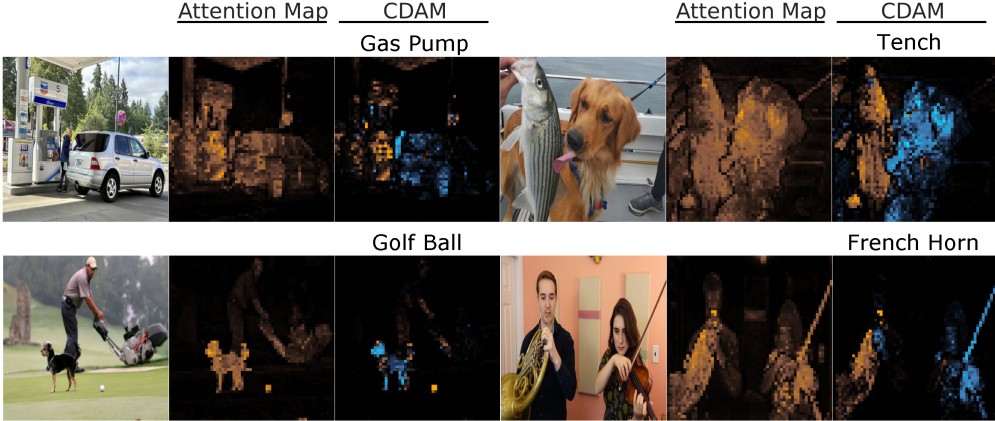

Figure 1: Extending attention maps (AM), the proposed class-discriminative attention maps (CDAM) quantify and visualize input features that are relevant for the target class in Vision Transformers (ViT). We visualize importance scores obtained with CDAM for a linear classifier with ViT-S/8, trained with DINO (Caron et al., 2021). Orange and blue colors correspond to positive and negative values, respectively.

# 1   Introduction

Vision transformers (ViT) are a family of computer vision models based on the transformer architecture (Vaswani et al., 2017), that demonstrate comparable or even better performance than convolutional neural networks (CNN) on many object detection and classification tasks (Dosovitskiy et al., 2020; Wu et al., 2020). A powerful feature of the transformers is that the attention mechanism provides a built-in explanation of the model's inner workings, if attention is understood as a weighing of the relevance of individual tokens (Mullenbach et al., 2018; Bahdanau et al., 2014; Serrano & Smith, 2019; Thorne et al., 2019). While *attention maps* provide intuitive and inherent interpretability, they fail to be discriminative with respect to a target class. Recognizing the potential of attention maps for explaining a ViT's inner workings, we have developed *class-discriminative attention maps* (CDAM), a gradient-based extension that makes them sensitive to class-specific signals based on a classifier or a latent concept.

Attention maps are the visualization of the self-attention of the class token query `[CLS]` in the final attention layer. Attention heads can be visualized separately, and in certain instances, may be related to semantically meaningful concepts (e.g., in language (Clark et al., 2019)). Recently, ViT trained with self-supervised learning (SSL) methods (Caron et al., 2021; Touvron et al., 2021; Oquab et al., 2024) have shown excellent performance leveraging a greater amount of unlabeled data. Attention maps from such self-supervised ViT models provide high-quality semantic segmentations of input images. These attention maps clearly distinguish regions of interest (ROI) from the background with very little noise, with different attention heads often focusing on semantically distinct regions (Caron et al., 2021). There have been many improvements to ViT including the introduction of convolution (Wu et al., 2021; Yuan et al., 2021) and spatial sparsity (Li et al., 2021). Attention maps, as well as our proposed CDAMs, are applicable to all ViT architectures with self-attention.

The major shortcoming of attention maps, when considering them as an explainable AI (XAI) method, is that they purely operate on the level of the last attention layer and therefore do not take into account any information coming from a downstream task, such as a classifier head. The proposed CDAM overcomes this shortcoming by calculating the gradients of the class score, i.e. the logit value of a target class, with respect to the token activations in the final transformer block (visualized in Fig. 2(a)). Essentially, CDAM scales the attention scores by how relevant the corresponding tokens are for the model's decision. CDAM therefore inherits the desirable properties of attention maps, but additionally indicates the evidence or counter-evidence for a target class that the ViT relies on for its predictions (examples in Fig. 1). We demonstrate that it is class-discriminative in the sense that the heat maps clearly distinguish the object representing the target class from the background and, importantly, from objects that belong to other classes. Therefore, CDAM is a useful tool for explaining the decision-making process of classifiers with a ViT backbone. Note that the goal of CDAM is to explain the classifier head and the final transformer block.

In addition to the classifier-based explanations, we present a concept-based approach that replaces the class score with the similarity measure in the latent space of the ViT, where a concept is defined through selected images. Averaging the latent representations of the example images yields a *concept embedding*, which is analogous to a concept vector in variational autoencoders (VAE) and related generative models (Brocki & Chung, 2019). This allows obtaining a heat map for the target concept the model has never been explicitly trained on, which is of particular interest for self-supervised models. We further demonstrate when computing gradients of a class (Section 3.1) or concept similarity (Section 3.2) score, with respect to the token activations in the final transformer block, smoothing or averaging could be applied to improve performance. Specifically, we introduce two additional variants of CDAM; (1) Smooth CDAM averages over multiple gradients that are computed with additional noise to tokens (inspired by SmoothGrad (Smilkov et al., 2017)) and (2) Integrated CDAM takes the integral of the gradients along the path from the baseline to the token in the final transformer block (inspired by IntGrad (Sundararajan et al., 2017)).

We conduct several quantitative evaluations focusing on correctness, class sensitivity, and compactness. Correctness is measured by perturbing an increasing amount of input features and measuring the model output (Brocki & Chung, 2023c;b). By using the ImageNet samples (Deng et al., 2009) with multiple objects (Beyer et al., 2020) and applying importance estimators for different classes, we quantify the level of class-discrimination. Sparsity and shrinkage of explanations are measured by the number of near-zero

importance scores. Besides the proposed vanilla, Smooth, and Integrated CDAM, we also employed 7 different explainability methods in comparison. Three variants of CDAM outperform other methods on quantitative benchmarks.

Lastly, we have applied CDAM on a ViT fine-tuned on the Lung Image Database Consortium image collection (LIDC) (Armato III et al., 2011). Two ViT-based models are built for predicting malignancy and biomarkers. CDAM is shown to be highly class-discriminative and semantically relevant, while providing implicit regularization of importance scores.

## 2  Related Works

Interpretability of ViT, and more broadly explainable artificial intelligence (XAI), is an active area of research, since understanding their decision-making process would help improve the model, identify weaknesses and biases, and ultimately increase human trust in them. The proposed class-discriminative attention map (CDAM) is related to both attention and gradient-based explanation methods for deep learning models.

Gradient-based methods operate by backpropagating gradients from the prediction score to the input features to produce feature attribution maps, which is also called saliency maps. Many variants have been proposed, see (Simonyan et al., 2013; Sundararajan et al., 2017; Selvaraju et al., 2017; Smilkov et al., 2017; Shrikumar et al., 2016) to just name a few. In particular, CDAM is similar to and inspired by the input×gradient method (Shrikumar et al., 2016), because it involves calculating the gradients with respect to tokens multiplied by their activation (Eqs. (2) and (4)). The main difference to most gradient-based methods is that we don't backpropagate the gradients all the way to the input tokens. CDAM stops backpropagation at the tokens that enter the final attention layer.

In that regard, our method is related to GradCam (Selvaraju et al., 2017) which backpropagates the gradients to the final convolutional feature map in a CNN. GradCam and CDAM therefore share the property that they operate on high-level features which is presumably the reason why they are both sensitive to the targeted class, a feature that many explanation methods do not have (Rudin, 2019). Several methods have attempted to apply gradient-based methods or Layer-Wise Relevance Propagation (LRP) (Bach et al., 2015) to Transformer architectures (Atanasova et al., 2020; Ali et al., 2022; Voita et al., 2019; Abnar & Zuidema, 2020; Chefer et al., 2021b), mostly in the context of NLP tasks. GradCam for ViT backpropages the gradients to the outputs of the final attention layer (Gildenblat, 2021).

Attention has often been used to gain insight into a model's inner workings (Mullenbach et al., 2018; Bahdanau et al., 2014; Serrano & Smith, 2019; Thorne et al., 2019) since it seems intuitive that this weighing of individual tokens correlates with how important they are for the performed task. There is also an ongoing discussion about whether attention provides meaningful explanations (Serrano & Smith, 2019; Wiegreffe & Pinter, 2019; Jain & Wallace, 2019). Furthermore, methods such as "attention rollout" (Abnar & Zuidema, 2020) have been proposed that aim to quantify how the attention flows through the model to the input tokens. (Chefer et al., 2021a) defines the relevancy matrices, as the Hadamard product of the attention map and the gradient of $f_c$ w.r.t. to the attention layer, accumulated at each layer. This relevance propagation (RP) method (Chefer et al., 2021a), to the best of our knowledge, shows the highest degree of class-discrimination in ViT models. This is used in our study as one of the baseline importance estimators.

Many papers have been published on the problem of evaluating post-hoc explanation methods, leading even to the proposal of meta-evaluations (Hedström et al., 2023). In a recent overview of quantitative evaluation methods (Nauta et al., 2023b) 12 desirable properties have been identified. We primarily focus on three of them, namely correctness, contrastivity, and compactness. In particular, carefully designed occlusion/perturbation studies can help us calculate the fidelity statistics (Brocki & Chung, 2023c), which is also called the symmetric relevance gain (SRG) measure (Bluecher et al., 2024). Adebayo et al. (2018) first demonstrated that many importance estimators, which purport to be sensitive to the internals of a classifier, produce similar importance scores regardless of the relation of inputs and classes in the training data, which they called "the data randomization test". Generally, if two importance scores with respect to two different classes produce very similar heat maps, the importance estimator is not class-sensitive. We quantify this

concept in our class sensitivity evaluation benchmark to measure whether an explanation is sensitive to the classes.

## 3 Methods

Adaptation of the transformer architecture to computer vision hinges on creating tokens from small patches of the input image (Dosovitskiy et al., 2020). In addition to those image patch tokens, the classification token [CLS] is added, which is utilized for downstream tasks such as classification. Attention maps are the two-dimensional visualization of the self-attention in the final attention layer with [CLS] as query token, where the attention scores are averaged over all heads. In our notation, [CLS] refers to the classification token before and [CLS]′ after the final transformer block.

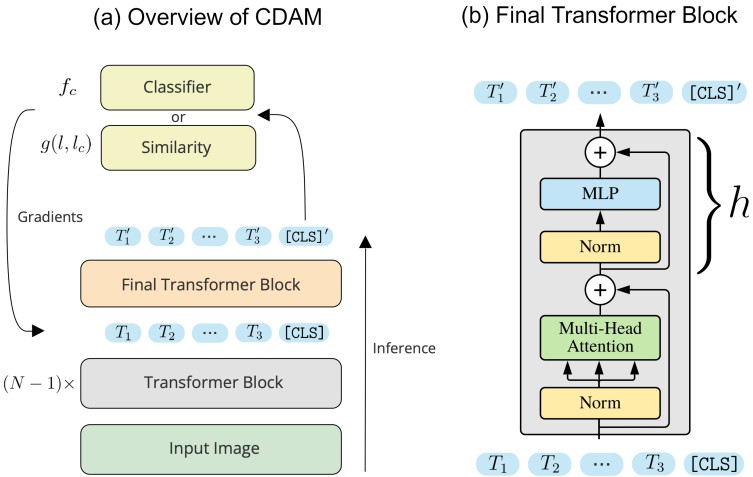

Figure 2: Graphical scheme of *class-discriminative attention maps* (CDAM). **(a)** CDAMs are obtained by first propagating an image through the transformer blocks and using its latent representation $l = $ [CLS]′ to infer a class score $f_c$ or similarity score $g(l, l_c)$ (see Eq. (2) or Eq. (4)). **(b)** Detailed view of the final transformer block, adapted from (Dosovitskiy et al., 2020).

Before fine-tuning, attention maps are not class-discriminative since they do not take into account any signal coming from a downstream task. Therefore, we have developed *class-discriminative attention maps (CDAM)* that can be computed based on a known class or latent concept. In order to estimate a token's relevance, CDAM computes gradients of a class score (Section 3.1) or concept similarity score (Section 3.2), obtained downstream using the [CLS]′ token, with respect to the token activations in the final transformer block (Fig. 2). Note that CDAM may not explain the full ViT architecture with multiple transformer blocks. Instead, CDAM focuses on the downstream task and the last transformer block.

In the first scenario, the class score is obtained from the corresponding activation in the prediction vector of a classifier trained on top of the ViT in a supervised fashion. In the second scenario, we define a concept through a small set of examples (e.g. ten images of a dog) and obtain a *concept vector* by averaging their latent representations. Then, without training a classifier, a similarity measure between the concept vector and the latent representation of a targeted sample image is computed and used as the target for the gradients. We call the resulting maps of feature relevance "class-discriminative attention maps" because they scale the attention scores

$$A_i = \text{softmax}\left( \frac{Q_{\texttt{[CLS]}} K_i^\top}{\sqrt{d_k}} \right), \tag{1}$$

by how relevant the corresponding tokens $T_i$ are for the class or similarity score. Here $Q_{\texttt{[CLS]}} = T_{\texttt{[CLS]}} W^Q$ with $W^Q \in \mathbb{R}^{d_{\text{model}} \times d_k}$ and $K_i = T_i W^K$ with $W^K \in \mathbb{R}^{d_{\text{model}} \times d_k}$. In this notation $Q$ and $K$ are the query

and key matrices which pack together all the query and key vectors (Vaswani et al., 2017) and the indices, e.g. $K_i$, select single query or key vectors.

Self-supervised learning (SSL) methods such as DINO can train ViT models whose attention maps are high-quality semantic segmentations of the input, without using labels (Caron et al., 2021). Therefore, we utilize a ViT model (Dosovitskiy et al., 2020) pre-trained with DINO as a backbone for our main experiments and applications. Furthermore, we demonstrate general applicability of CDAM by using other ViT models, namely pre-trained with Supervised Weakly through Hashtags (SWAG) (Singh et al., 2022), Data-efficient image Transformers (DeIT) (Touvron et al., 2021) and DINOv2 (Oquab et al., 2024). We denote details about the ViT architecture, training schemes, and model performance in the Section 4 and Appendix A.5. Note that the heat maps shown in this article are scaled independently.

### 3.1  CDAM for a known class

Consider a linear classifier $f : \mathbb{R}^{d_{\text{model}}} \rightarrow \mathbb{R}^C$ on top of the ViT that takes the `[CLS]`$'$ token as input and maps it to a prediction vector with $C$ classes, where $d_{\text{model}}$ is the embedding dimension of the ViT. The importance score $S_{i,c}$ with respect to the token $T_i$ is defined as

$$S_{i,c} = \sum_j T_{ij} \frac{\partial f_c}{\partial T_{ij}}, \tag{2}$$

where $T_{ij}$ is the $j$-th component of the $i$-th token that is fed to the last attention layer (Fig. 2). We found in our experiments that calculating the gradients with respect to the layer-normalized tokens yields far better results than using the not normalized ones.

This definition is a first-order estimation of the contribution of $T_i$ to $f_c$ in the sense that it would be exact if $f_c$ depended on $T_{ij}$ only linearly. It is inspired by gradient-based explanation methods with the difference that we don't backpropagate the gradients all the way to the input layer but only to the input tokens of the final attention layer. This allows us to gain insight into the decision-making process of the combination of ViT and classifier based on high-level features. This approach leads to attention maps that are class-discriminative and well-aligned with the semantic relationships of the class and different parts of an image (Fig. 1).

### 3.2  CDAM for a latent concept

Another way of probing high-level features learned by the ViT is through the use of a similarity measure $g$ in the latent space of the ViT. We compare the latent representation $l(X) = T_{\text{[CLS]}'}(X)$ of a sample image $X$ and a *concept vector* $l_c$ (Brocki & Chung, 2019), i.e. the averaged latent representation of $n$ images that are chosen to represent the concept of interest:

$$l_c = \frac{1}{n} \sum_{X \in c} l(X). \tag{3}$$

This approach is directly inspired by the concept saliency map (Brocki & Chung, 2019). The definition of the importance score in this setting is analogous to Eq. (2)

$$S_{i,c} = \sum_j T_{ij} \frac{\partial g(l, l_c)}{\partial T_{ij}}. \tag{4}$$

Possible choices for $g$ are for example the $L^2$ distance, cosine similarity or dot product, where we use the latter in this article. The definition Eq. (4) estimates the contribution of $T_i$ to $g$, such that it allows us to gauge how much each token is driving the similarity between $l$ and $l_c$.

### 3.3  Smooth and Integrated CDAM

SmoothGrad (SG) (Smilkov et al., 2017) and Integrated Gradients (IG) (Sundararajan et al., 2017) extended saliency maps (Simonyan et al., 2013), which were introduced in the context of CNNs as visualizing gradients

of the class with respect to the input features. Briefly, (Smilkov et al., 2017) adds a small Gaussian noise to a sample image $n$ times, resulting in $n$ vanilla saliency maps. SG is an average of those $n$ saliency maps. In IG, instead of adding random noise, (Sundararajan et al., 2017) uses multiple saliency maps obtained from interpolating between a given sample and a baseline (e.g., a black image).

We have adapted the analogous modifications to CDAM, resulting in two additional variants which we call Smooth and Integrated CDAM. Importantly, instead of modifying the input images, we vary the tokens that enter the final transformer block. In Integrated CDAM, we interpolate between a baseline and the tokens themselves, resulting in $n$ maps. Notice that, since we apply the same method, the axioms of IG still hold with the only difference that they apply to the tokens that enter the final transformer block instead of the input pixels. In Smooth CDAM, we repeatedly add a small Gaussian noise to the tokens and again obtain $n$ maps.

Specifically, Smooth and Integrated CDAM are defined as follows:

$$S_{i,c}^{\text{SG}} = \frac{1}{n} \sum_{1}^{n} \sum_{j} T_{ij}' \frac{\partial f_c(T')}{\partial T_{ij}'} \tag{5}$$

$$S_{i,c}^{\text{IG}} = \sum_{j} (T_{ij} - \tilde{T}_{ij}) \times \frac{1}{n} \sum_{1}^{n} \frac{\partial f_c(\tilde{T} + k/n \times (T - \tilde{T}))}{\partial T_{ij}} \tag{6}$$

where $T_i' = T_i + \mathcal{N}\left(0, \sigma^2\right)$ with $\mathcal{N}\left(0, \sigma^2\right)$ representing Gaussian noise with standard deviation $\sigma$. For our experiments, we used $n = 50$. For the baseline $\tilde{T}_i$ we choose the null vector of 0. We implement these two variants by first propagating an image through the ViT up to the final transformer block, where the activations of the tokens are obtained. Next, the activations are manipulated by either applying the noise or interpolation step and the manipulated tokens are then further propagated to obtain the final ViT output.

### 3.4 Quantitative Evaluation

We perform quantitative evaluations of the correctness, class sensitivity, and compactness of importance estimators. In addition to our proposed methods (vanilla, Smooth, and Integrated CDAM), we perform evaluations on attention maps, Input × Gradients, relevance propagation (RP) (Chefer et al., 2021a), partial Layer-wise Relevance Propagation (LRP) (Voita et al., 2019), SmoothGrad (Smilkov et al., 2017), IntGrad (Sundararajan et al., 2017), and GradCam for ViT (Gildenblat, 2021). Note that based on classic LRP (Bach et al., 2015), partial LRP (Voita et al., 2019) provides improvement for multi-head self-attention mechanisms in modern transformer architecture (Voita et al., 2019). Relevance propagation is a state-of-the-art importance estimator for explaining a prediction (e.g., class discriminatory) based on ViT (Chefer et al., 2021a). For IntGrad, SmoothGrad, Integrated CDAM, and Smooth CDAM, $n = 50$ is used. The noise ($\sigma$) used in SmoothGrad and Smooth CDAM is selected based on the best performance on a validation set of 200 images.

To evaluate the correctness (also known as faithfulness) of importance scores, we employ two versions of input perturbation-based approaches (Samek et al., 2016; Petsiuk et al., 2018; Kindermans et al., 2017). Generally, an input feature (e.g., a pixel) is masked and a resulting model performance is observed. In all cases, input features are perturbed by being replaced with their blurred versions. A Gaussian blur with $\sigma = 14$ turns pixels uninformative while minimizing perturbation artifacts (Brocki & Chung, 2023c).

**Fidelity.** The first evaluation method uses accuracy curves obtained from the Most Important First (MIF) and the Least Important First (LIF) perturbations, which are abbreviated as MIF/LIF perturbation (Brocki & Chung, 2023c;b). Pixels are ranked by importance scores from a specific explainability method (e.g., CDAM) and are perturbed in the MIF or LIF order. In the MIF perturbation, an accurate importance score should lead to a relatively rapid decrease of the model output (logit) for the target class. For the LIF perturbation, we expect an inverse relationship. Note that some importance scores are unsigned (i.e., only

non-negative values) and others are signed. For signed importance, with negative scores faithfully indicating counter-evidence, there may be an increase in the logits if pixels with negative scores are masked. For unsigned and correct importance scores, one expects initially constant logits in the LIF perturbation.

The model output $f_{\mathrm{MIF}}(p)$ or $f_{\mathrm{LIF}}(p)$ as a function of the fraction of perturbed pixels $p$ gives the perturbation curve (Brocki & Chung, 2023c;b). We average these curves over all samples, providing an overview of fidelity for a given importance estimator. We further summarize the MIF and LIF perturbation evaluation by using the area $A_{\mathrm{LIF\text{-}MIF}}$ (called the fidelity statistics) under the difference of perturbation curves $f_{\mathrm{LIF}} - f_{\mathrm{MIF}}$ to quantify the fidelity of the explanation methods (Brocki & Chung, 2023b). Generally speaking, a higher value of fidelity statistics $A_{\mathrm{LIF\text{-}MIF}}$ indicates a better explanation (Brocki & Chung, 2023c). Similar ideas are developed into the symmetric relevance gain (SRG) measure (Bluecher et al., 2024).

**Box sensitivity.** For the second evaluation method, inspired by Sensitivity-n (Ancona et al., 2017), we measure how the model output (logit) changes as a group of pixels is perturbed. Pixels are selected by a randomly placed square box (of size $s \times s$); then, the sum of their importance scores and the change in the model output are measured. For accurate importance estimators, we expect to see a high correlation between the two measures.

For a given image, we sample these values 100 times and calculate the Pearson correlation between the sum of importance scores and the model output change. This is performed for a range of differently sized boxes to obtain $f_{\mathrm{box}}(s)$, the correlation statistics as a function of the box size $s$. These sensitivity curves are averaged over all samples. Performance of importance estimators varies with $s$, and therefore we use the area $A_{\mathrm{box}}$ under the box sensitivity curve to quantify correctness. Generally, a larger box sensitivity statistics $A_{\mathrm{box}}$ implies greater correctness.

In contrast to *Sensitivity-n* which randomly selects and perturbs $n$ pixels (Ancona et al., 2017), we use a random square of pixels. A box removes a certain region which mimics how image patches are processed by ViT. Box sensitivity may therefore remove larger parts or complete objects. In contrast, removing random pixels may leave enough neighboring pixels with sufficient semantic information such that perturbation may not have any impact of the model output. Indeed, we have observed that for Sensitivity-n, the correlation is very low ($< 0.1$) for all methods.

**Class discrimination.** We have designed an evaluation benchmark to measure class discrimination, which can be applied on images containing multiple objects (see $X_{\mathrm{multi}}$ dataset below). We obtain explanations for two distinct classes corresponding to the present objects. When performing the aforementioned evaluation benchmarks (fidelity and box sensitivity), we measure the change in the model output with respect to one of the classes. Evaluation metrics are thus obtained using importance scores based on either the correct target class (denoted by MIF/LIF or box) or a wrong class (denoted by MIF′/LIF′ or box′).

For important estimators that are not sensitive/discriminative to the target class, the importance scores are similar (or the same) regardless of which class is targeted. The change of the model output during perturbation would remain the same, irrespective of the class that was targeted to obtain the importance ranking of pixels. For highly class-discriminative importance estimators, importance scores would be different depending on which class is targeted. Furthermore, for accurate class-discriminative methods, the performances on fidelity and box sensitivity benchmarks can be expected to deteriorate substantially when perturbing according to the importance scores targeting the wrong class.

The areas under MIF/LIF perturbation or box sensitivity curves, when using a wrong class, are denoted by $A_{\mathrm{LIF'-MIF'}}$ and $A_{\mathrm{box'}}$, respectively. The areas $\Delta A_{\mathrm{LIF\text{-}MIF}}$ and $\Delta A_{\mathrm{box}}$ under the respective perturbation curves

$$\Delta f_{\mathrm{LIF\text{-}MIF}} = (f_{\mathrm{LIF}} - f_{\mathrm{MIF}}) - (f_{\mathrm{LIF'}} - f_{\mathrm{MIF'}}), \tag{7}$$

$$\Delta f_{\mathrm{box}} = f_{\mathrm{box}} - f_{\mathrm{box'}}. \tag{8}$$

measure the degree of class-discrimination. In particular, if an importance estimator is not sensitive to the target class at all, $\Delta f_{\mathrm{LIF\text{-}MIF}} \approx \Delta f_{\mathrm{box}} \approx 0$. An importance estimator that performs poorly in the data randomization test of (Adebayo et al., 2018) would have low $\Delta A_{\mathrm{LIF\text{-}MIF}}$ and $\Delta A_{\mathrm{box}}$.

**Compactness.** We evaluate sparsity and shrinkage by counting the number of input pixels with importance scores smaller than 5% of the maximum value. A threshold is defined by $t \times S_c^{\max}$ with $t = 0.05$, where $S_c^{\max}$ is the maximum importance score for a given image and an importance estimator. Generally, a compact explanation is considered desirable, where sparsity and shrinkage are quantitative approaches (Nauta et al., 2023a).

### 3.5 Data

We have used two subsets of the ImageNet (denoted as $X_{\text{val}}$ and $X_{\text{multi}}$) (Deng et al., 2009), as well as lung computed tomography (CT) scans from the Lung Image Database Consortium image collection (LIDC) (Armato III et al., 2011). The first one, $X_{\text{val}}$, is a randomly selected subset of the ImageNet validation set consisting of 1000 images (Deng et al., 2009). The second one, $X_{\text{multi}}$, selects 1000 images from the validation set that contain multiple objects (Beyer et al., 2020). We filtered out instances in which the class labels provided by (Beyer et al., 2020) actually refer to the same object (e.g. notebook and laptop) by calculating and thresholding the cosine similarity of the embedded class names. Embeddings were obtained using OpenAI's *text-embedding-3-large* model.

The $X_{\text{multi}}$ dataset is used to evaluate the degree of class-discrimination exhibited by the explanations. To this end, we obtain explanations w.r.t to two annotated classes, perform the MIF/LIF perturbation and box sensitivity, and record the model output for the chosen target class. For example, for the top right sample in Fig. 1, we would obtain the explanations for the classes *Tench* and *Golden Retriever* (not shown) and record the model output for the class *Golden Retriever*. For the right target (*Tench*), we obtain a fidelity statistics $A_{\text{LIF}-\text{MIF}}$. For the wrong target (*Golden Retriever*), $A_{\text{LIF}'-\text{MIF}'}$. In the worst case scenario, where explanations for those two distinct classes are the same, $A_{\text{LIF}'-\text{MIF}'} = A_{\text{LIF}-\text{MIF}}$ and the resulting $\Delta A_{\text{LIF-MIF}}$ is zero. Equivalent for $\Delta A_{\text{box}}$.

The Lung Image Database Consortium image collection (LIDC) contains 1018 records of lung CT scans, that were collected from and validated by seven academic centers and eight medical imaging companies (Armato III et al., 2011). Presence of nodules (nodule $\leq$ 3 mm; nodule $<$ 3 mm; non-nodule $\geq$ 3 mm) is annotated by up to 4 human annotators. Segmentation of nodules (i.e., ROI) is the average contour given by annotators. Eight biomarkers that are informative in clinical practices are also collected in LIDC: subtlety, calcification, sphericity, margin, lobulation, spiculation, texture, and diameter. For clinical descriptions of biomarkers, refer to (Opulencia et al., 2011). After preprocessing, 854 nodule crops of size, corresponding biomarkers, and labels (benign or malignant) were saved and used for downstream procedures. Overall, the LIDC dataset used in this study contains 443 benign examples and 411 malignant examples.

## 4 Results

For our experiments using the ImageNet (Deng et al., 2009), we use the ViT-S/8 architecture (with a patch size of $8 \times 8$) trained with DINO as a backbone (Caron et al., 2021). Instead of using a ViT model that is pre-trained in a self-supervised manner, we demonstrate CDAMs directly from an alternative ViT architecture trained with supervised weakly through hashtags (SWAG) (Singh et al., 2022) in Appendix A.5. We further demonstrate applicability of CDAM on computed tomography (CT) scans, using DINO (Caron et al., 2021) in Section 5, as well as using Data-efficient image Transformers (DeIT) (Touvron et al., 2021) and DINOv2 (Oquab et al., 2024) in Appendix A.5.2.

### 4.1 Qualitative Evaluation

#### 4.1.1 CDAM for a known class

To demonstrate our method, we have trained a linear classifier on the ImageNet (Deng et al., 2009) that achieved an accuracy of 77.0%. We use random resized cropping and horizontal flipping with PyTorch default arguments as augmentation, Adam optimizer with learning rate $3 \times 10^{-4}$, batch size of 128, and train for 10 epochs. The parameters of the ViT backbone are frozen during training, so only the classifier head is

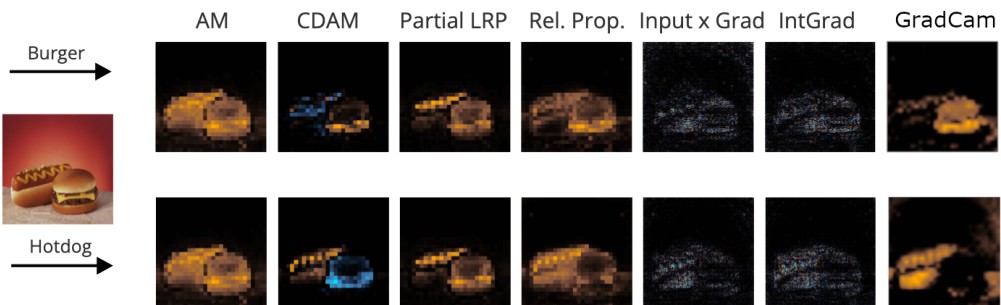

Figure 3: Visual comparison of different importance estimators which obtain explanations w.r.t. different output classes. Similar heat maps regardless of target classes imply that importance estimators are not class discriminative. Attention maps (AM) are, by design, not class discriminative. Orange and blue colors correspond to positive and negative values, respectively.

trainable. The predictions of this classifier were then used to create the heat maps shown in Fig. 1 via Eq. (2).

CDAM clearly distinguishes the region of interest (ROI) semantically related to the targeted class with positive values (yellow) from the rest of the image. Most backgrounds and other objects are assigned either zero (irrelevant) or negative scores (counter-evidence). Note that CDAM does not assign non-zero $S_{i,c}$ to tokens that have zero attention in attention maps, see also the scatter plot between CDAM and attention map Fig. S1. CDAM therefore appears to compute $S_{i,c}$ by multiplying the attention with the relevance of the corresponding token for the targeted class. We discuss the proportionality of CDAM to the attention scores in Appendix A.1. It is a nice feature since CDAM inherits the high-quality object segmentations present in attention maps. This qualitative evaluation of shrinkage and sparsity of CDAM is reflected in the quantitative measures for compactness (Section 4.2.1).

One of the most useful features of CDAM is that it is highly sensitive to the chosen class. In Fig. 5 (left), when either the *zuccini* or *bell pepper* class is chosen, CDAM clearly highlights that vegetable. Most of the other vegetables are assigned either near-zero or negative values (indicated in blue). Many other importance estimators do not distinguish the target class or only to a lesser degree. In Fig. 3, targeting the burger or the hotdog does not seem to change importance scores from partial LRP, Input × Grad, and IntGrad. Relevance propagation demonstrates some selective focus on the ROI, but is not as discriminative as CDAM (Fig. 3). This visual impression of class-sensitivity is supported by the quantitative evaluation in Section 4.2.2. Additional examples of CDAM and other interpretability methods are shown in Appendix A.3.

### 4.1.2    CDAM for a latent concept

Concept embeddings $l_c$ have been obtained by averaging latent representations of 30 images. Those images were randomly selected from a class in the ImageNet validation set (Deng et al., 2009). Then, CDAMs in Fig. 4 were obtained using Eq. (4). We observe that CDAM reveals that the model clearly distinguishes the parts of the images that are relevant for the latent concepts shared by selected images from the rest of the image. For example, when having selected images containing *hammer* (bottom right), CDAM primarily highlights the hammer. While the annotated classes were used for this experiment, the concepts shared by those images are likely more nuanced, specific, and complex than the class itself. For example, CDAM also provides positive importance scores for the wooden handles of the screwdrivers and the socket attached the ratchet, which imply relevance for the concepts shared by 30 images selected from the class *hammer*. Semantically, 30 images that are selected from a class *hammer* have a wooden handle (i.e., concept), which is shared by multiple tools. Similarly, the socket at the top of the ratchet shares similarities with typical metal heads at the top of the hammer. Unrelated tools or parts appear in blue indicating that the model deems them to be unrelated to the concept *hammer*.

CDAMs are discriminative with respect to the target concepts shared by selected samples (Fig. 5, right). This makes the proposed method particularly valuable for understanding the representations the ViT has learned,

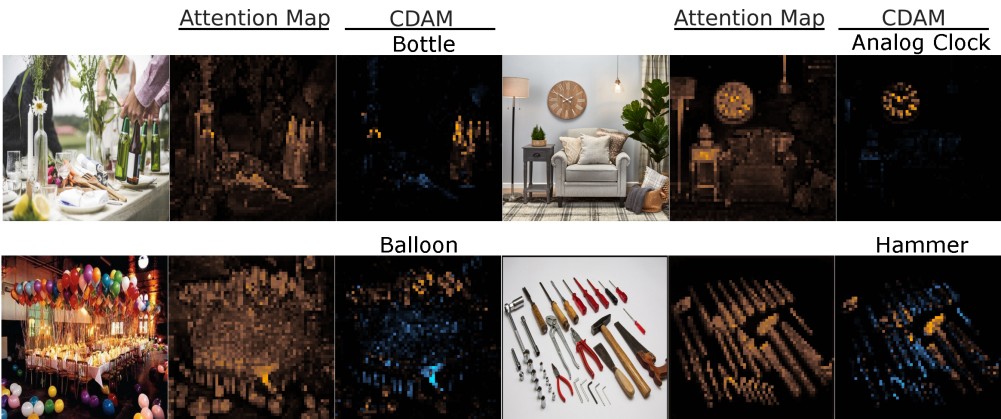

Figure 4: Class-Discriminative Attention Maps (CDAM) for user-defined concepts (Section 3.2). The concept embedding $l_c$ has been obtained by averaging the latent representations of 30 images that include the common concept. In each instance, from left to right: Sample image, AM, and CDAM. Orange and blue correspond to positive and negative values, respectively.

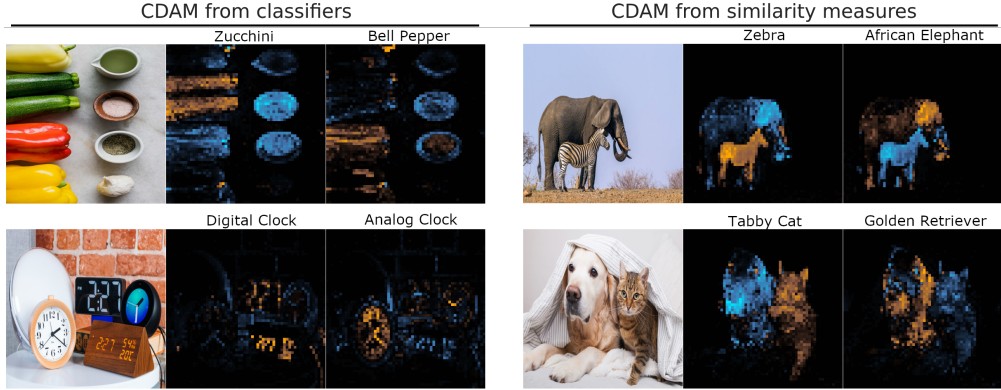

Figure 5: When targeting different classes (left) or concepts (right), the resulting CDAMs are distinct and align with the corresponding objects. Note that green parts of vegetables are shown to activate the *zuccini* class (left). The wrinkles in the elephant's trunk appear to be related to the *zebra* concept, semantically mistaken by the model as zebra stripes.

since the explanations for different concepts can give a complementary view. For example, by selecting 30 images that contain *african elephant*, CDAM highlights areas that are important for concepts which coincide largely with the body of an elephant. It also shows that parts of the elephant's trunk are counter-evidence for the concepts shared by images with *african elephant* (Fig. 5, right). For the same sample image, we can also select 30 images with *zebra*, which suggests that the model deems the part of the elephant's trunk to be similar to concepts shared by those images.

Although we selected a group of images based on annotated classes, which are denoted in Fig. 5 (right), to illustrate concept-based CDAMs, generally the concepts shared by selected images are not identical to the classes themselves. One could manually provide a set of images with a shared concept that is not originally annotated in the dataset. For example, 10 images that contain *zebras* may share semantic concepts (e.g., "stripes" and "tail"), that make up the class *zebra*. We demonstrate this by selecting a set of 10 images, outside of the ImageNet, that exclusively show *stripes* and another set of 10 images that contain diverse animal *tails*. Shown in Fig. 6, CDAM for *stripes* highlights the body of the zebra, but not the body of the cheetah. On the other hand, CDAM for *tails* tends to focus on the tail of the cheetah. Note that 10 images of diverse animal *tails* include body parts of animals, often rear ends and legs.

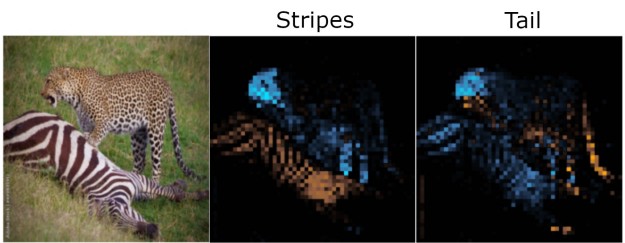

Figure 6: Concept-based CDAM based on 10 images that show *stripes* or contain animal *tails*.

## 4.2 Quantitative evaluation

We have performed quantitative evaluations of the correctness and compactness of our proposed methods, in comparison to multiple well-known importance estimators. We further investigated class sensitivity by using importance scores from multiple classes.

### 4.2.1 Evaluation of correctness

|  | $A_{\text{MIF}}\downarrow$ | $A_{\text{LIF}}\uparrow$ | Fidelity $A_{\text{LIF-MIF}}\uparrow$ | Box Sensitivity $A_{\text{box}}\uparrow$ |
|---|---|---|---|---|
| CDAM | 404 | 1038 | 632 | 12.2 |
| Integrated CDAM | 375 | 1045 | 669 | **13.0** |
| Smooth CDAM | 282 | **1110** | **827** | 12.4 |
| Relevance Propagation | 315 | 957 | 643 | 10.0 |
| Partial LRP | 489 | 837 | 348 | 4.5 |
| Attention map | 452 | 846 | 393 | 4.8 |
| SmoothGrad | 534 | 529 | -4 | 0.4 |
| IntGrad | **226** | 802 | 574 | 7.6 |
| GradCAM | 300 | 944 | 643 | 8.6 |
| Input $\times$ Grad | 273 | 713 | 439 | 3.5 |

Table 1: Evaluation of correctness by fidelity and box sensitivity. In $X_{\text{multi}}$, importance scores for the target class were used to perturb pixels. The changes in the model output w.r.t. the perturbation percentage ($f_{\text{LIF-MIF}}$; Fig. S2) or the box size ($f_{\text{box}}$; Fig. S3) are quantified by the area under the curves.

CDAM performs favorably in terms of the correctness of estimated feature importance compared to other methods. For the dataset containing multiple objects (Table 3), the Smooth CDAM outperforms all other methods on the fidelity evaluation using MIF/LIF perturbation (Fig. S2), with $A_{\text{LIF-MIF}} = 827$ (Table 3). On the box sensitivity benchmark, Integrated CDAM ($A_{\text{box}} = 13.0$) performs the best, closely followed by Smooth CDAM ($A_{\text{box}} = 12.4$) (Fig. S3). Some methods, such as IntGrad, perform very well on the MIF benchmark but poorly for LIF (Table 3). We consider the difference (LIF-MIF) to be more meaningful and intuitive as ideally, one would want a large area under the LIF perturbation curve and a small area under the MIF perturbation curve. Thus, larger $A_{\text{LIF-MIF}} \uparrow$ indicates higher correctness. Particularly, a good score in MIF could stem from simply triggering perturbation artifacts of the model (Brocki & Chung, 2023b; Hooker et al., 2019).

For the random subset of ImageNet, the Integrated CDAM demonstrates the best performance ($A_{\text{box}} = 12.9$), closely followed by Relevance Propagation ($A_{\text{box}} = 12.7$) on the box sensitivity benchmark (Fig. S6, Table S1). Relevance Propagation outperforms all other methods on the MIF/LIF perturbation, followed by the Smooth CDAM (Fig. S7).

|  | Fidelity $A_{\Delta(\text{LIF-MIF})}\uparrow$ | Box Sensitivity $A_{\Delta\text{box}}\uparrow$ |
|---|---|---|
| CDAM | 739 | 14.6 |
| Integrated CDAM | 708 | **14.8** |
| Smooth CDAM | **823** | 13.4 |
| Relevance Propagation | 293 | 6.6 |
| Partial LRP | 3 | 0.0 |
| Attention map | 0 | -0.2 |
| SmoothGrad | -11 | -0.1 |
| IntGrad | 540 | 7.6 |
| GradCAM | 681 | 9.4 |
| Input $\times$ Grad | 433 | 3.4 |

Table 2: Evaluation of class discrimination by differences in fidelity and box sensitivity metrics. By using the $X_{\text{multi}}$ dataset, we measure how sensitive importance scores are to changing their target class (Details in Section 3.5). Area under the $\Delta f_{\text{LIF-MIF}}$ (Fig. S4) and $\Delta f_{\text{box}}$ curves (Fig. S5).

### 4.2.2 Evaluation of class discrimination

Class discrimination (or sensitivity) is measured by the difference in fidelity statistics $A_{\Delta(\text{LIF-MIF})}$ when perturbing according to importance scores obtained for the correct or wrong target class. Larger $A_{\Delta(\text{LIF-MIF})}$ is considered to indicate better performance.

CDAM ($A_{\Delta(\text{LIF-MIF})}$=739), Smooth CDAM ($A_{\Delta(\text{LIF-MIF})}$=823), and Integrated CDAM ($A_{\Delta(\text{LIF-MIF})}$=708) clearly outperform other methods in terms of class discrimination (Table 3). IntGrad (540), Input $\times$ Grad (433), and Relevance Propagation (293) follow. Surprisingly, Partial LRP and SmoothGrad perform deficiently. Whereas SmoothGrad has low class-discrimination because its importance scores have low accuracy to begin with, Partial LRP's scores have mediocre accuracy but it appears to be insensitive to the choice of the target class (Fig. 3). By design, attention maps do not consider the target class and therefore, $A_{\Delta(\text{LIF-MIF})}$=0.

The importance scores for CDAM become anti-correlated with the model output when the *wrong* importance scores are used for the perturbation (Fig. S5). This anti-correlation probably results from the fact that pixels that are evidence for the correct target class are counter-evidence for the wrong class. This result corroborates the visual impression that CDAM correctly assigns importance scores with opposite signs to the objects corresponding to the targeted class and other objects that are present (Fig. 3). This is also reflected in the negative values of $f_{\text{LIF}'} - f_{\text{MIF}'}$ for CDAM (Fig. S4), indicating that the ranking from least to most important has been, at least partially, reversed.

### 4.2.3 Evaluation of compactness

Sparsity and shrinkage are evaluated by our compactness evaluation, which is one of the major desired properties in the explainability of deep learning (Nauta et al., 2023a). While sparsity may not always imply the most accurate estimation at a single data point, a bias-variance trade-off is well-known in machine learning (Lahlou Kitane, 2022). Ideally, compact explanations (e.g., sparse and shrunken importance estimators) are preferred, when other properties such as correctness and class sensitivity are constant.

CDAM and Integrated CDAM resulted in the highest degree of compactness, together with Input $\times$ Grad. For those three importance estimators, on average, 88% of importance scores were less than 5% of the maximum score Table 3. IntGrad and SmoothGrad show relatively high sparsity 0.84 and 80%, respectively. But both are lacking in correctness (Section 4.2.1) and class-discrimination (Section 4.2.2). Relevance Propagation shows the lowest sparsity of all considered methods 48%, but a high level of correctness. These results suggest that these correctness and compactness metrics are orthogonal.

|  | Compactness↑ |
|---|---|
| CDAM | **0.88** |
| Integrated CDAM | **0.88** |
| Smooth CDAM | 0.75 |
| Relevance Propagation | 0.48 |
| Partial LRP | 0.74 |
| Attention map | 0.56 |
| SmoothGrad | 0.8 |
| IntGrad | 0.84 |
| GradCAM | 0.54 |
| Input × Grad | **0.88** |

Table 3: Compactness, measuring sparsity and shrinkage, is defined by the fraction of pixels with an importance score lower or equal to 5% of the maximum score, evaluated on the $X_{\mathrm{multi}}$ dataset.

## 5 Application to Medical Images

To demonstrate CDAM for another use case we apply it to nodule malignancy and biomarker prediction using the Lung CT scans in the LIDC dataset (Armato III et al., 2011). Here, we show CDAMs from DINO-based models, we also showcase use of DeIT (Dosovitskiy et al., 2020; Touvron et al., 2021) and DINOv2 (Oquab et al., 2024; Darcet et al., 2024) backbones that are fine-tuned on the LIDC dataset in Appendix A.5.2.

### 5.1 Malignancy Prediction

After preprocessing, the LIDC data consists of 443 benign and 411 malignant lung CT scans. Training, validation, and test sets (in the ratios of 0.7225, 0.1275, 0.15) were stratified by and balanced according to these labels, e.g., benign and malignant. We fine-tuned a ViT model, with a DINO backbone pre-trained on the ImageNet, for 50 epochs. In a parameter sweep, we varied the number of trainable layers $(10 - 50)$ and dropout rates $(0.0 - 0.09)$, where the learning rate was exponentially decaying ($\alpha = 0.0003$ and $\beta = 0.95$). The best accuracy on the test set of 0.85 was obtained with 50 trainable layers and dropout rate of 0.003[1]. Attention maps and CDAM are obtained based on this model.

The obtained attention maps suggest that the model focuses on patches with nodule fragments (Fig. S11(b)). However, we note that attention maps almost always focus on pulmonary nodules without taking into account the downstream classification into benign and malignant samples. CDAM provides detailed structures where positive and negative values are indicated by orange and blue, respectively. In this binary classification where 0 is for benign and 1 for malignancy, input pixels with positive values in CDAMs are driving the classification towards malignancy.

### 5.2 Biomarker prediction

We turn our attention to clinical biomarkers in LIDC that are routinely used by medical practitioners. Inspired by the concept bottleneck model (CBM) (Koh et al., 2020), we fine-tuned a ViT model pre-trained with DINO on 8 biomarkers (subtlety, calcification, sphericity, margin, lobulation, spiculation, texture, and diameter). Essentially, a regression model was built on a pre-trained ViT model with the mean squared error (MSE) as loss function. For investigation of LIDC and incorporation of CBM, see (Brocki & Chung, 2023a).

CDAMs were obtained for 8 biomarkers (Fig. 7). In the context of the margin, patches including the edge of a nodule get positive CDAM scores, whereas patches corresponding to the nodule body get negative CDAM scores. For the diameter, patches corresponding to the nodule body get a positive score, whereas the background pixels exhibit neutral (near zero) or slightly negative scores. When it comes to sphericity, patches including fragments of nodule curvature get a positive CDAM score. Spiculation refers to the presence of

---

[1]It might be possible to increase the model performance via further investigation. Here, we use a straightforward model and training scheme to demonstrate the application of CDAM on medical images.

needle and spike-like structures, on the margin. High spiculation often is an indication of malignancy. In a majority of cases, CDAM highlights spicules with positive values, whereas the inner bodies of nodules (i.e., not spicules) tend to get low importance scores.

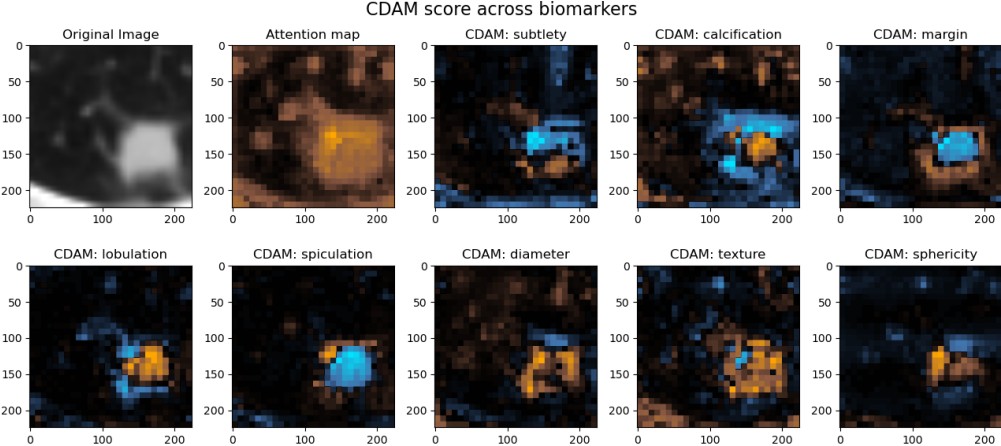

Figure 7: Explanations for the biomarker prediction model. Eight biomarkers available in the LIDC dataset are used to fine-tune a ViT regression model with a DINO backbone. CDAMs provide distinct and relevant explanations for different biomarkers. For more examples, see Appendix A.4.

## 6 Discussion

Class-Discriminative Attention Map (CDAM) is a gradient-based extension of attention maps, that provides strong discrimination with respect to a chosen class or concept, while exhibiting excellent fidelity and compactness. Our proposed methods retain appealing characteristics of attention maps, namely their high-quality semantic segmentation. The proposed importance estimation for CDAM scales the attention scores by how relevant they are for a target class or concept (Appendix A.1). Therefore, zero attention scores remain zero in CDAM and it produces compact explanations with high sparsity and shrinkage. Besides explaining predictions of the classifier on the top of ViT, CDAM can also provide importance scores specific to the user-defined concept. In the context of self-supervised models, where class labels are absent, this makes it a valuable technique to investigate the latent representations learned by the model.

Post-hoc explanation methods for DNNs aim to make the full decision-making process of the model more transparent by providing an approximation with certain desirable properties, such as correctness, sensitivity, and compactness. Particularly, many explanation methods that estimate feature importance are not very useful because they give almost identical results when targeting different classes in the model output (Rudin, 2019; Adebayo et al., 2018). Even randomization in the model weights is often shown to have minimal impact on explanations (i.e., saliency maps (Adebayo et al., 2018)). We find, both qualitatively and quantitatively, that CDAM is highly sensitive to the targeted class, assigning positive importance scores to objects corresponding to the target class and negative ones to other (semantically distinct) objects in the image.

We introduce Smooth and Integrated CDAM, which essentially average a series of (vanilla) CDAMs. Notably, instead of adding noise to or modifying the input images as in SmoothGrad (Smilkov et al., 2017) or IntGrad (Sundararajan et al., 2017), respectively, our methods act on tokens in the final transformer block. While we conducted a number of quantitative and qualitative evaluations, there is no single XAI method that triumph in all scenarios. However, considering that Smooth and Integrated CDAM require computing $n$ vanilla CDAMs (e.g., $n = 50$), we recommend first trying vanilla CDAMs. If high fidelity is singularly desirable, we suggest utilizing Smooth CDAM.

Our application on medical images exemplifies the need for fine-grained class-sensitive explanations offered by CDAM. Other explainability methods including attention maps often highlight whole nodules and tumors. However, they are uninformative about the inner workings of the ViT model predicting malignancy or

biomarkers. As the methods of explainable AI (XAI) are considered for practical implementations, the investigation of human interactions with different explainability methods represents a promising direction for future studies.

**Broader Impact Statement**

The introduction of our importance estimators would enhance the transparency, trustworthiness and accountability of vision transformer models. By providing clearer insights into model decision-making processes, accurate explainability methods help foster broader adoption of AI technologies. However, the explanations could be manipulated by adversarial input perturbation or data poisoning. In turn, enhanced explanations may inadvertently expose vulnerabilities, making models more susceptible to adversarial attacks or malicious exploitation. Therefore, it is imperative to continuously monitor and ensure the robustness of explanation methods while implementing safeguards to mitigate the risks associated with adversarial attacks.

**Acknowledgments**

This work was partially funded by the SONATA BIS grant [2023/50/E/ST6/00694] from the National Science Centre of Poland (Narodowe Centrum Nauki). This work was partially funded by the ERA-Net CHIST-ERA grant [CHIST-ERA-19-XAI-007] long term challenges in ICT project INFORM (ID: 93603), by the National Science Centre (NCN) of Poland [2020/02/Y/ST6/00071]. This research was carried out with the support of the Interdisciplinary Centre for Mathematical and Computational Modelling University of Warsaw (ICM UW) under computational allocation no GDM-3540; the IDUB program (Excellence Initiative - Research University), the NVIDIA Corporation's Academic Hardware Grant; and the Google Cloud Research Innovators program.

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

## A    Appendix

### A.1    Proportionality of CDAM to attention maps

The proposed class-discriminative attention scores scale the attention scores. Empirically, in all of our evaluation tasks and applications (e.g., Fig. S1), we observe that features with attention scores $A_i = \text{softmax}\left(\frac{Q_{\text{[CLS]}}K_i^\top}{\sqrt{d_k}}\right)$ of 0 have class-discriminative attention scores of 0:

$$A_i = 0 \Rightarrow S_{i,c} = 0. \tag{9}$$

This operating characteristic is highly favorable, as besides introducing class discrimination, CDAM provides implicit regularization resulting in sparser heat maps. Based on a combination of empirical observations and mathematical derivations, we argue that the attention score $A_i = \text{softmax}\left(\frac{Q_{\text{[CLS]}}K_i^\top}{\sqrt{d_k}}\right)$ is scaled by the relevance of the corresponding token $T_i$ for a concept or a class.

To better understand this observation, we first consider a concept-based CDAM with a single attention head. Eq. (4) is expanded:

$$S_{i,c} = \sum_j T_{ij} \frac{\partial g(l, l_c)}{\partial T_{ij}} \tag{10}$$

$$= \sum_j T_{ij} \frac{\partial}{\partial T_{ij}} \sum_k l_k l_{c,k} \tag{11}$$

$$= \sum_j T_{ij} \frac{\partial}{\partial T_{ij}} \sum_k h_k \left( \sum_m A_m V_m + \text{[CLS]} \right) l_{c,k} \tag{12}$$

$$= \sum_j T_{ij} \sum_{k,m,n} \nabla^n h_k \times \left( \frac{\partial A_m}{\partial T_{ij}} V_{mn} + A_m \frac{\partial V_{mn}}{\partial T_{ij}} \right) l_{c,k} \tag{13}$$

$$= \sum_{j,k,m,n} T_{ij} \nabla^n h_k \frac{\partial A_m}{\partial T_{ij}} V_{mn} l_{c,k} + \sum_{k,n} A_i V_{in} \nabla^n h_k l_{c,k}, \tag{14}$$

where we assume the architecture of (Dosovitskiy et al., 2020) and use the projection of the tokens onto the key vector $V_i = T_i W^V$ with $W^V \in \mathbb{R}^{d_{\text{model}} \times d_v}$. We have also used the fact that the latent representation $l$ (i.e. $\text{[CLS]}'$) is the sum of value vectors weighted by the corresponding attention, $l = \sum_i A_i V_i$ with $A_i = \text{softmax}\left(\frac{Q_{\text{[CLS]}}K_i^\top}{\sqrt{d_k}}\right)$ plus $\text{[CLS]}$ due to the residual connection, see Fig. 2(b). The function $h : \mathbb{R}^{d_v} \to \mathbb{R}^{d_{\text{model}}}$ consists of the layer normalization, MLP, and residual connection and performs the final processing of the $\text{[CLS]}$ token before it enters the classifier (Fig. 2(b)). $\nabla^n$ is the $n$-th component of the gradient and $Q_{\text{[CLS]}}$, $V_i$ and $K_i$ are the rows of matrices $Q, V$ and $K$ that correspond to the $\text{[CLS]}$ and the i-th tokens.

Eq. (9) implies that in Eq. (10), all terms that are not proportional to $A_i$ are zero or cancelled out and we are left with

$$S_{i,c} = A_i V_i \cdot \nabla(h \cdot l_c). \tag{15}$$

We thus find that $S_{i,c}$ is obtained by multiplying $A_i$ with the directional derivative of $h \cdot l_c$ in the direction of $V_i$. In other words, CDAM scales $A_i$ by the rate at which the dot product (similarity) between $h$ and $c$ changes in the direction of $V_i$.

In the special case that $h$ is the identity function, we obtain an even simpler interpretation of CDAM due to $\nabla_n h_k(f) = \partial_{f_n} f_k = \delta_{nk}$. Assuming $\frac{\partial A_m}{\partial T_{ij}} = 0$, Eq. (10) simplifies to $S_{i,c} = A_i V_i \cdot l_c$, which means that $A_i$ is scaled by the similarity of $V_i$ and $l_c$.

We note that introducing $h$ attention heads does not change the above derivations qualitatively. Instead of the sum $Z = \sum_m A_m V_m$, there will be a term $\text{Concat}(Z^1, Z^2, .., Z^h)W^O, W^O \in \mathbb{R}^{hd_v \times d_{\text{model}}}$ so that in the

case of multi-head attention the equivalent of Eq. (10) reads

$$S_{i,c} = \text{Concat}\left(Y^1, Y^2, .., Y^h\right) W^O \nabla(h \cdot l_c), \ Y^n = T_i \sum_m \frac{\partial A_m^n}{\partial T_i} V_m^n + A_i^n V_i^n \tag{16}$$

and since $A_i^n > 0, A_i = \sum_n \frac{A_i^n}{h}$ implies that $A^i = 0 \Rightarrow A_i^n = 0, \forall n$ we can again conclude that in order for Eq. (9) to hold all terms that are not proportional to $A_i$ either vanish or cancel out. For multi-head attention, we thus arrive at

$$S_{i,c} = \text{Concat}\left(Y'^1, Y'^2, .., Y'^h\right) W^O \nabla(h \cdot l_c), \ Y'^n = A_i^n V_i^n, \tag{17}$$

which does not have such a straightforward interpretation as the single-headed case due to the mixing of the attention heads.

If we think of $W^O$ as acting first on $\nabla(h \cdot l_c)$, creating a column vector of dimension $hd_v$, we can understand Eq. (17) as the sum of $h$ directional derivatives of the $d_v$-long segments of this columns vector in the directions of $Y'^1$ to $Y'^h$. Each of the summands is proportional to the corresponding $A^n$, and the interpretation of CDAM from the single-head case therefore extends to the multi-head one.

Although our observation $A_i = 0 \Rightarrow S_{i,c} = 0$ implies that all terms that are not proportional to $A_i$ in Eq. (10) should vanish, we don't have a theoretical argument why that would be the case. It seems unlikely that the various terms in the first sum cancel out, which leaves $\frac{\partial A_m}{\partial T_{ij}} = 0$ as an explanation. However, in general, $A_i$ depends on $T_i$ through $K_i$ and one would therefore expect the derivative to be non-zero.

We can also show that CDAM scales the attention by the relevance of the corresponding token for a chosen class (instead of concept), in the case that a classifier with weights $W^C$ has been trained on top of the ViT. Analogously to the above single-head discussion we have

$$S_{i,c} = \sum_j T_{ij} \frac{\partial f_c}{\partial T_{ij}} \tag{18}$$

$$= \sum_j T_{ij} \frac{\partial}{\partial T_{ij}} \sum_k l_k W_{kc}^C \tag{19}$$

$$= \sum_{j,k,m,n} T_{ij} \nabla^n h_k \frac{\partial A_m}{\partial T_{ij}} V_{mn} W_{kc}^C + \sum_{k,n} A_i V_{in} \nabla^n h_k W_{kc}^C, \tag{20}$$

and the only difference is that the directional derivative now acts on $\sum_k h_k W_{kc}$, i.e. the contribution of $h$ to the activation of class $c$ in the prediction vector, instead of $\sum_k h_k l_{c,k}$. All the intuitions we have gained from the concept-based CDAM therefore extend to the class-based one.

## A.2  Quantitative Evaluations

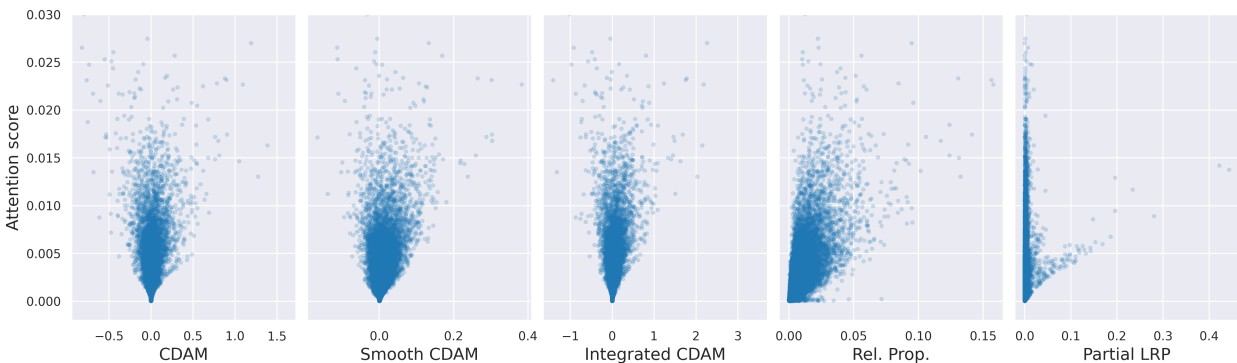

Figure S1: Scatter plot of raw attention scores and token importance scores obtained from 200 randomly selected samples of the ImageNet validation set $X_{\mathrm{val}}$.

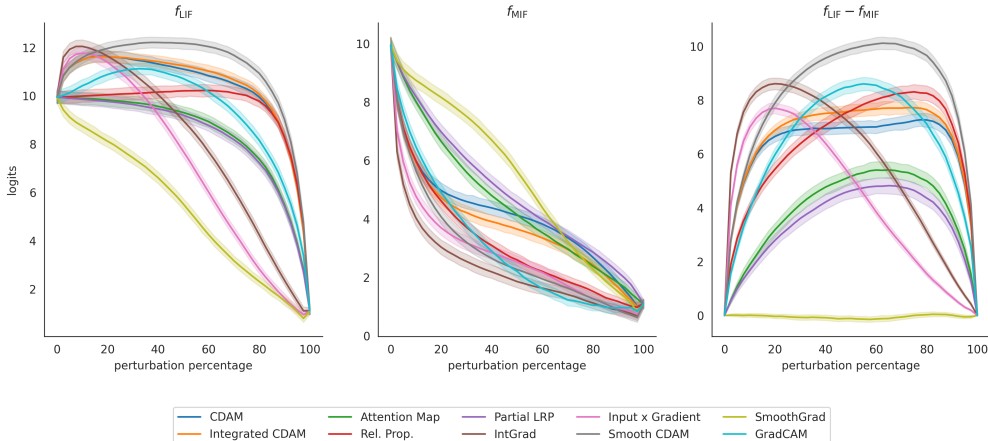

Figure S2: Fidelity of importance estimators based on MIF/LIF perturbation curves obtained from the $X_{\mathrm{multi}}$ dataset. The change in the model output is measured as a proportion of input features are perturbed according to the Least Important First (LIF) or the Most Important First (MIF) orders. Importance estimators are evaluated by the area between LIF and MIF perturbation curves, which is equivalent to the area underneath $f_{\mathrm{LIF}}$-$f_{\mathrm{MIF}}$ (right). See Table 3.

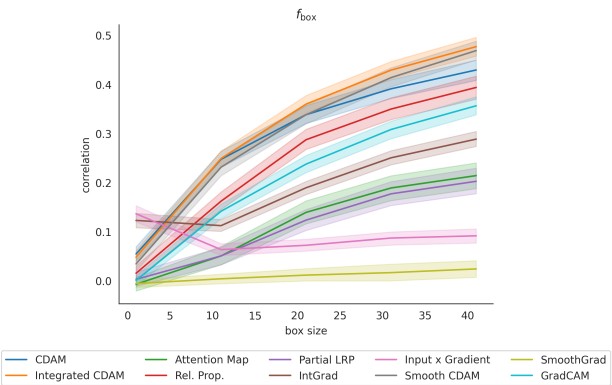

Figure S3: Correctness of importance estimators as measured by box sensitivity evaluation. The change in the model output (logit) is measured as pixels within a $s \times s$ box are perturbed. The sum of their importance scores would correlate with the change in the model output, if importance estimators are accurate. This is repeated 100 times per sample and the Pearson correlation between the sum of importance scores and the change in the model output is calculated, based on the $X_{\mathrm{multi}}$ dataset.

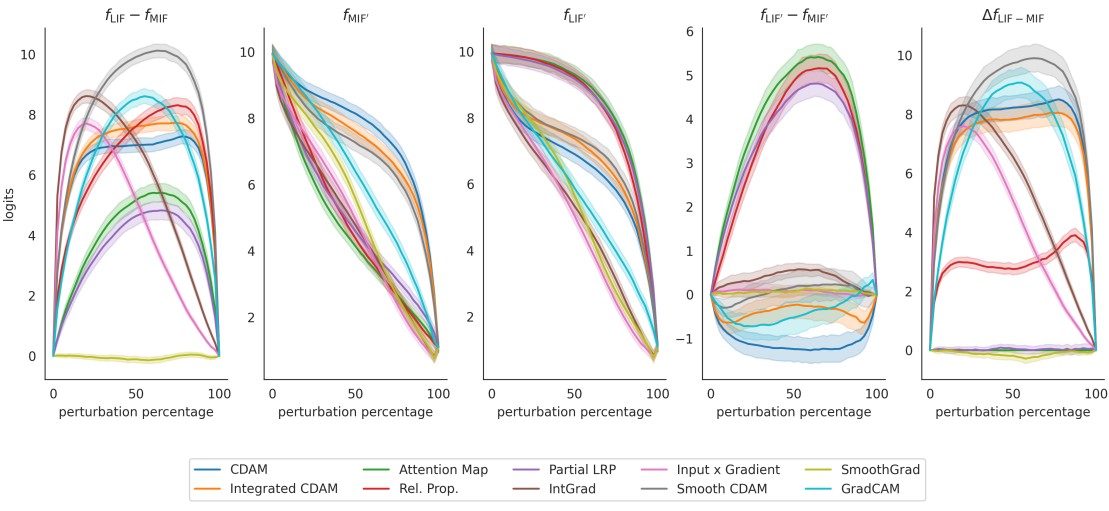

Figure S4: MIF/LIF perturbation curves for the evaluation of class-discrimination. Different classes are targeted for obtaining importance scores and measuring model output (Section 3.5), a large $\Delta A_{\mathrm{box}}$, therefore, indicates importance scores that are class-discriminative. See Table 2

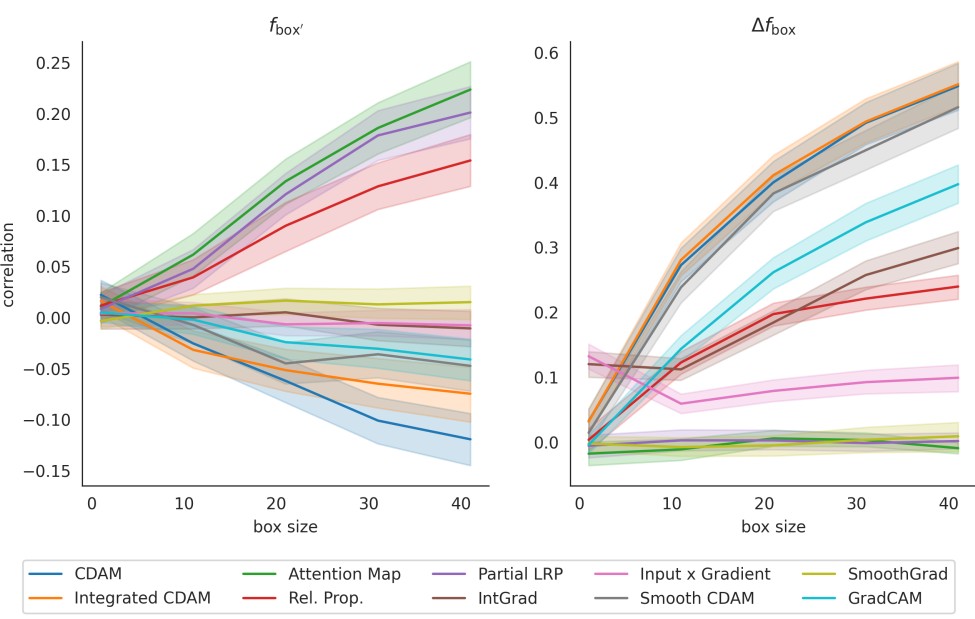

Figure S5: Box sensitivity curves for the evaluation of class-sensitivity

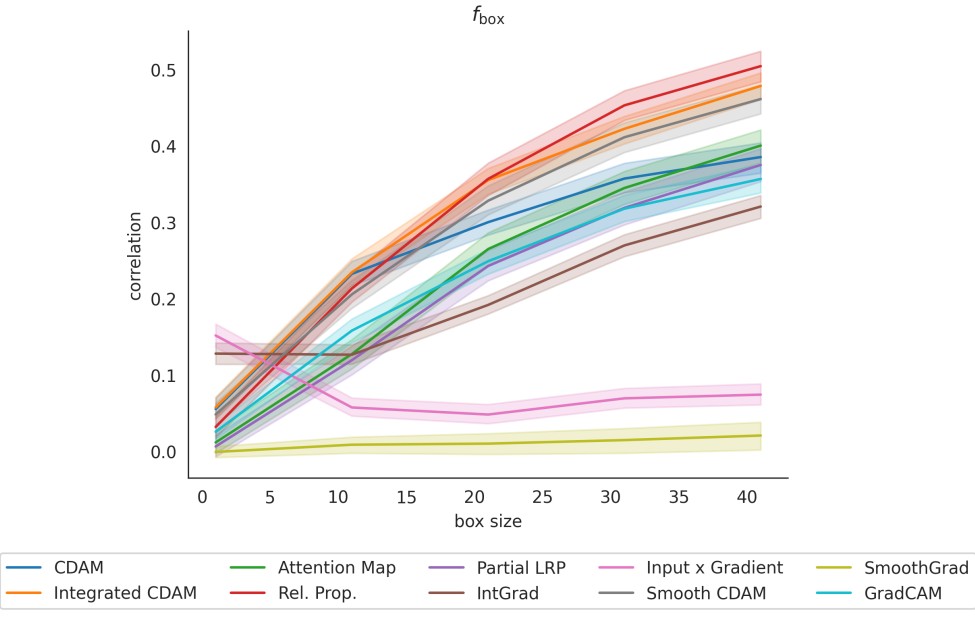

Figure S6: Box sensitivity curves obtained from the random ImageNet subset $X_{\mathrm{val}}$. Refer to Section 3.5 for data description.

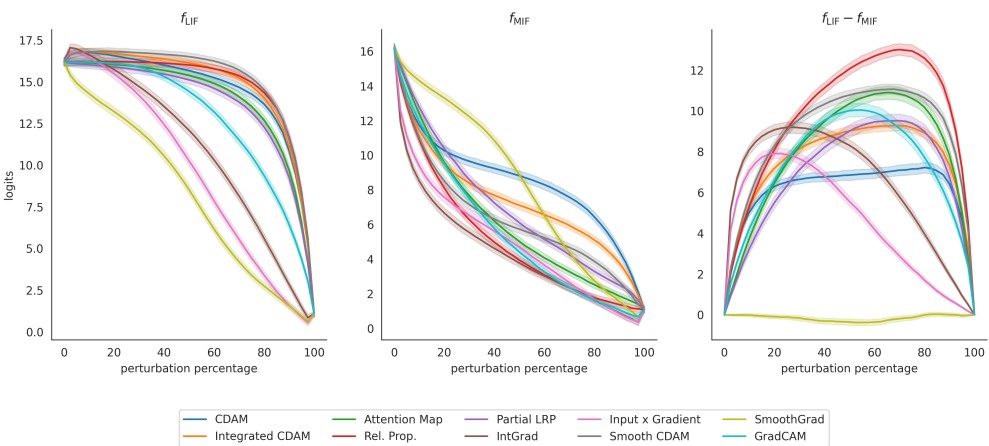

Figure S7: MIF/LIF perturbation curves obtained from the random ImageNet subset $X_{\mathrm{val}}$. Refer to Section 3.5 for data description.

|  | $A_{\mathrm{MIF}}\downarrow$ | $A_{\mathrm{LIF}}\uparrow$ | $A_{\mathrm{LIF\text{-}MIF}}\uparrow$ | $A_{\mathrm{box}}\uparrow$ |
|---|---|---|---|---|
| CDAM | 801 | 1423 | 623 | 11.3 |
| Integrated CDAM | 696 | 1459 | 753 | **12.9** |
| Smooth CDAM | 591 | **1482** | 890 | 11.9 |
| Rel. Prop. | **475** | 1432 | **957** | 12.7 |
| Partial LRP | 654 | 1329 | 674 | 8.4 |
| Att. map | 580 | 1352 | 771 | 9.1 |
| SmoothGrad | 796 | 786 | -10 | 0.2 |
| IntGrad | 411 | 1055 | 644 | 8.3 |
| GradCAM | 515 | 1249 | 733 | 9.0 |
| Input x Grad | 470 | 935 | 465 | 3.1 |

Table S1: Area under the curve for MIF, LIF and LIF-MIF curves (Fig. S7) and box sensitivity curves (Fig. S6) obtained from the random ImageNet subset.

### A.3 Examples of CDAMs from the ImageNet

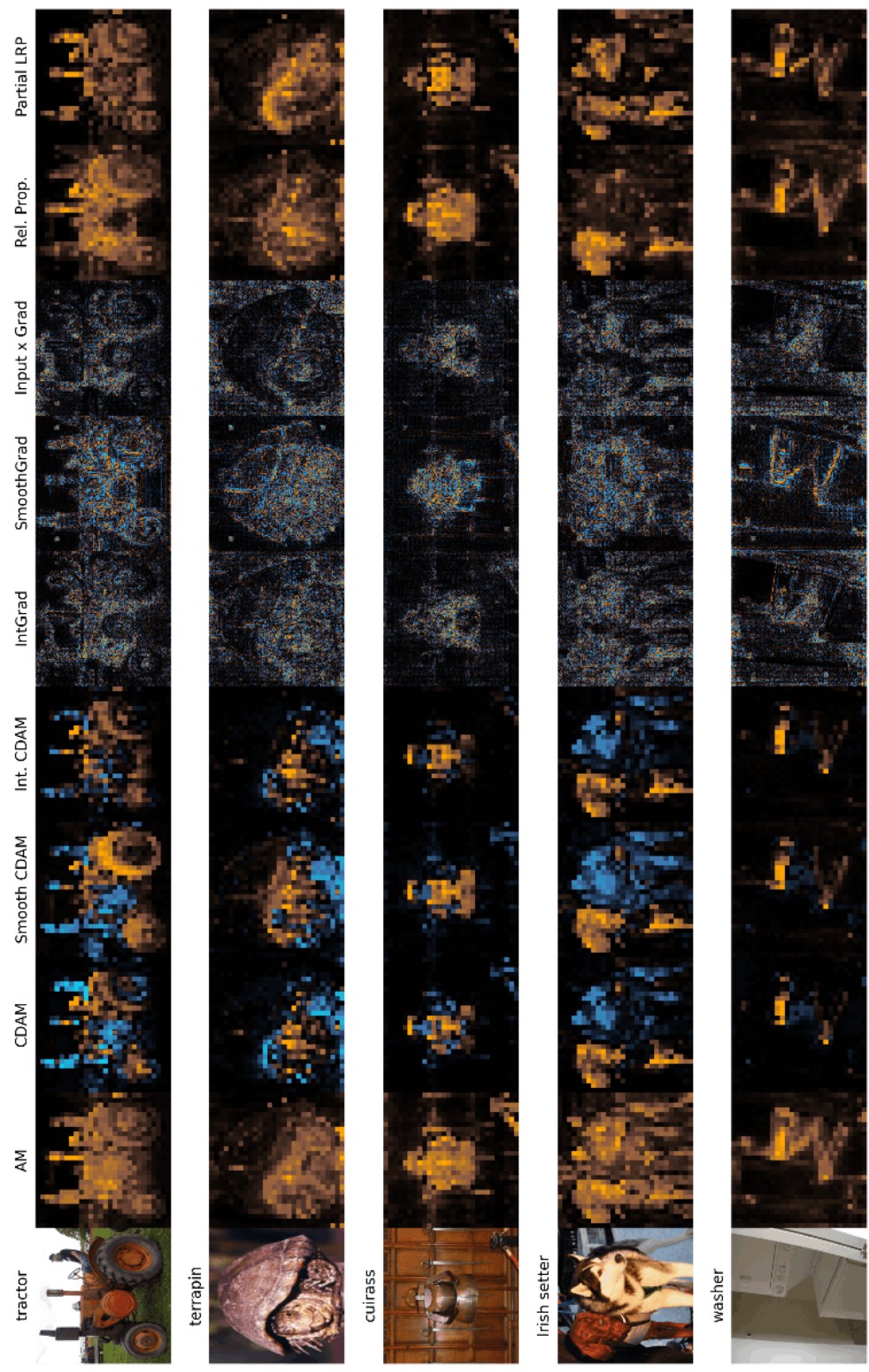

Figure S8: Additional examples for CDAM and other explanation methods on the ImageNet. The indicated class is the one predicted by the model and is the target for the explanation methods.

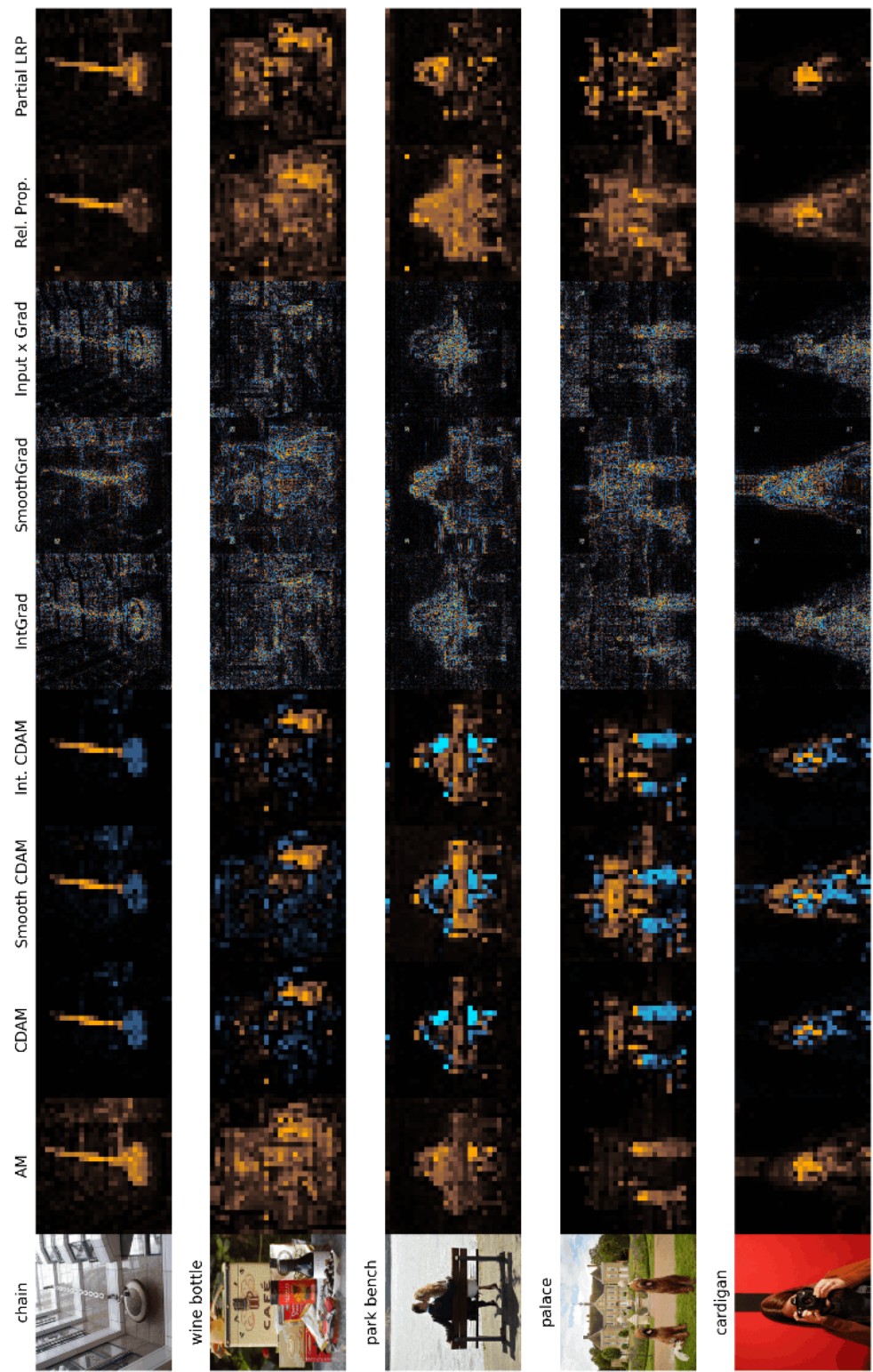

Figure S9: Additional examples for CDAM and other explanation methods on the ImageNet. The indicated class is the one predicted by the model and is the target for the explanation methods.

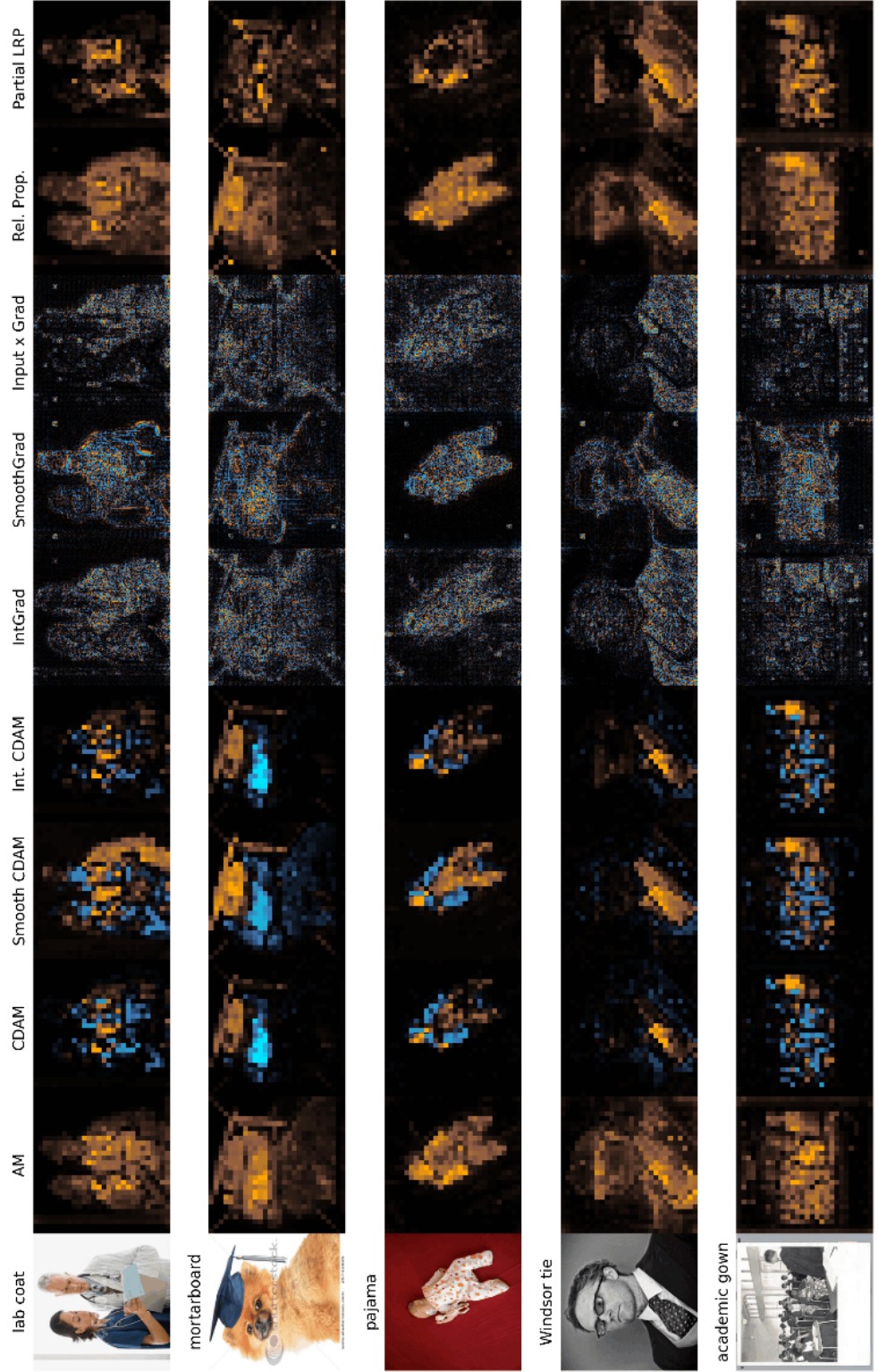

Figure S10: Additional examples for CDAM and other explanation methods on the ImageNet. The indicated class is the one predicted by the model and is the target for the explanation methods.

### A.4   Examples for CDAMs from lung CT scans

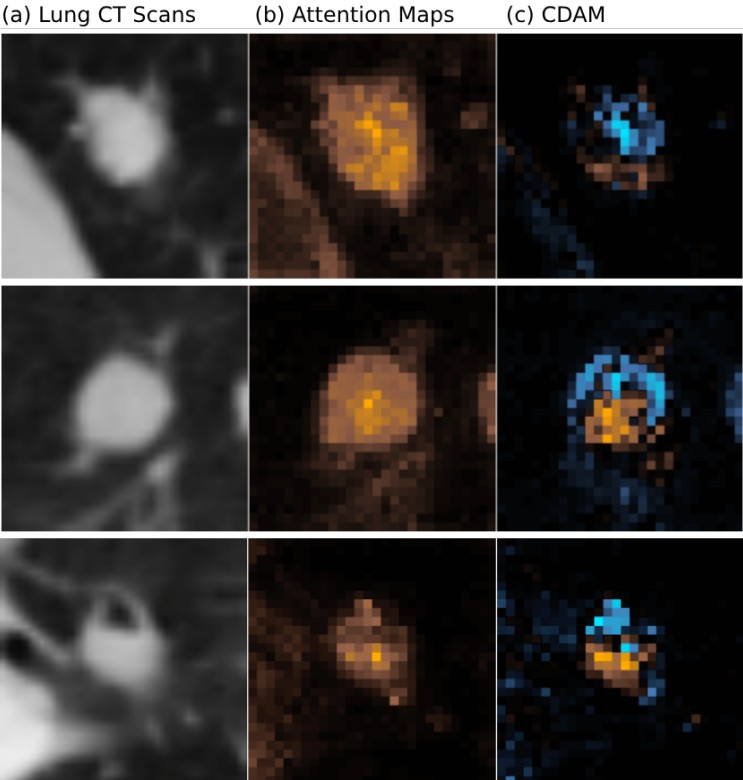

Figure S11: Explanations for the malignancy prediction model. A ViT-based classifier fine-tuned on the lung CT scans (LIDC) was used to predict benign vs. malignant lung nodules, followed by obtaining their attention maps and CDAMs. Orange and blue correspond to positive and negative values, respectively.

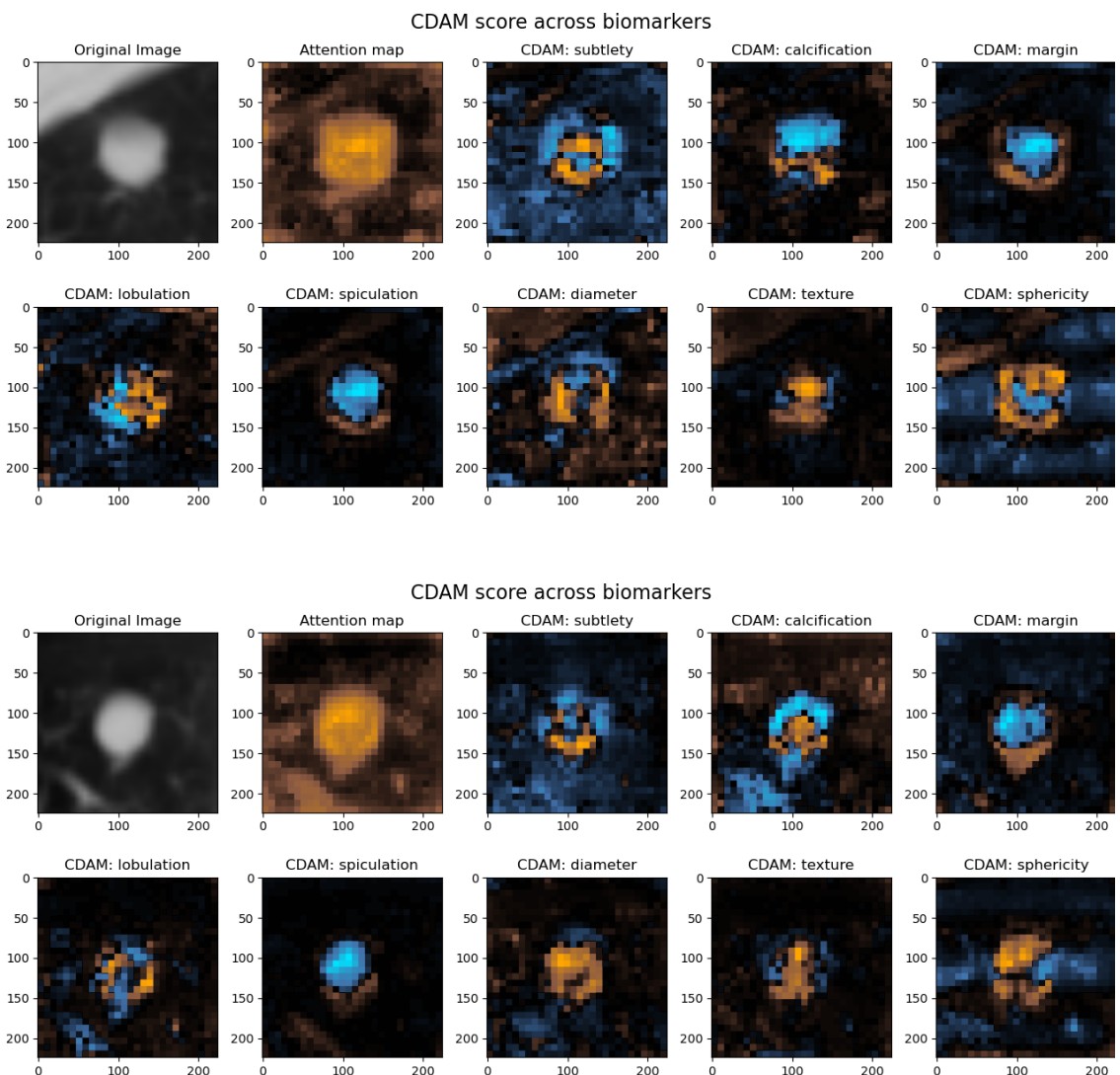

Figure S12: Examples for the Attention Map and CDAM based on a ViT biomarker model trained on the lung nodule data from LIDC.

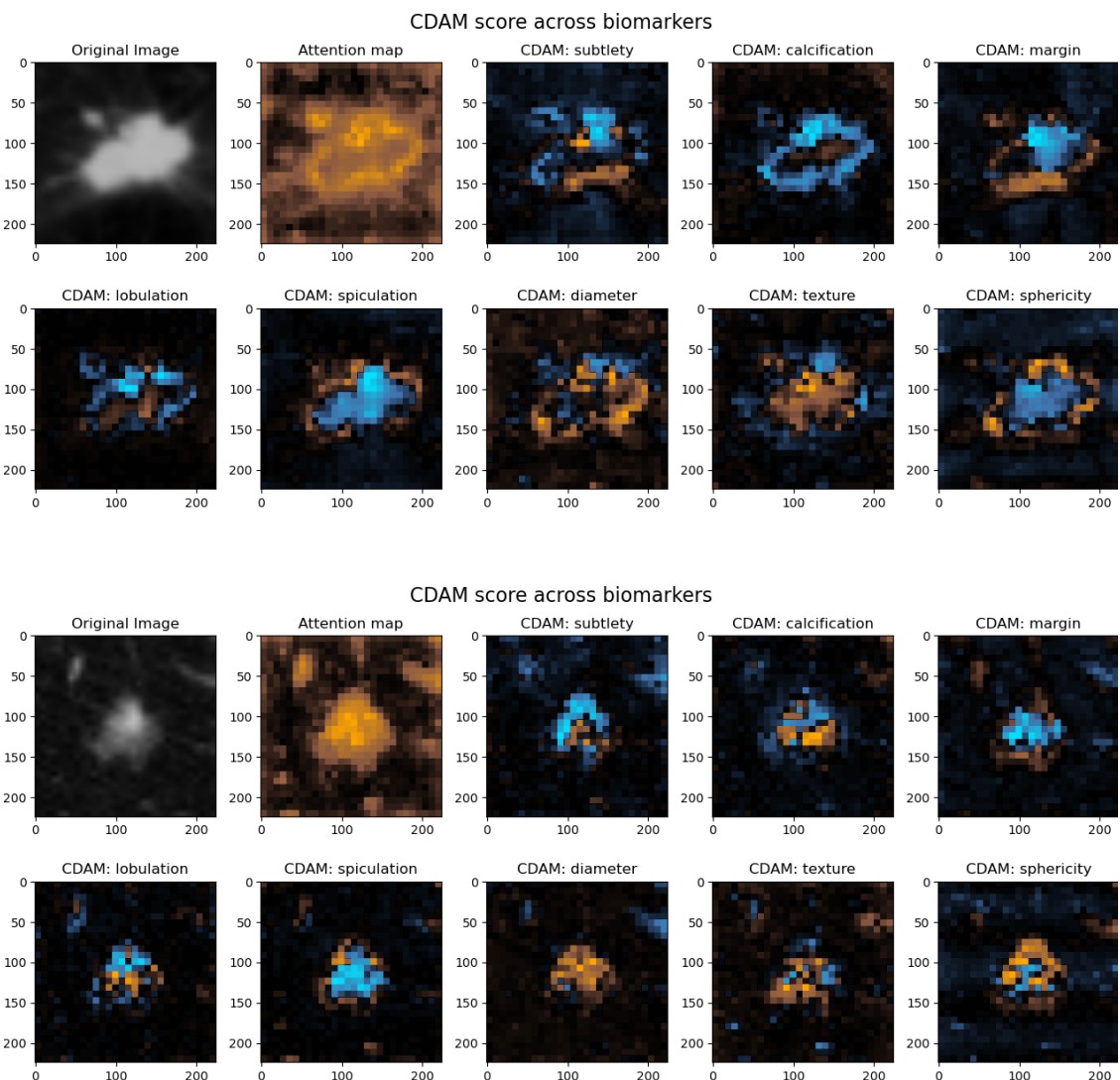

Figure S13: Examples for the Attention Map and CDAM based on a ViT biomarker model trained on the lung nodule data from LIDC.

### A.5 Additional ViT architectures and training strategies

While the main manuscript has used ViT models pre-trained using DINO (Caron et al., 2021), other architectures and pre-trained weights can be used for CDAM. In this section, we show examples of alternative ViT models that are trained in alternative manners. As expected, pre-trained ViT models that produce high-quality attention maps result in high-quality CDAMs. In particular, note that the patch size of DINO (Caron et al., 2021) is $8 \times 8$ compared to much larger patch sizes used in other backbones.

Table S2: Pre-trained ViT models and training strategies.

| Name | Model sizes | Patch sizes | References |
|---|---|---|---|
| DeIT (original ViT) | 86M | 16 | Dosovitskiy et al. (2020); Touvron et al. (2021) |
| SWAG | 86M | 16 | Singh et al. (2022) |
| DINOv2 (with registers) | 21M, 86M | 14 | Oquab et al. (2024); Darcet et al. (2024) |
| DINO (used in the main text) | 21M, 85M | 8, 16 | Caron et al. (2021) |

### A.5.1 Alternative ViT models for the ImageNet

We use a ViT model pre-trained on the ImageNet-1K dataset using supervised weakly through hashtags (SWAG) (Singh et al., 2022). Unlike DINO and other self-supervised ViT models, the SWAG model has been trained end-to-end. Therefore, without pre-training, we can obtain attention maps and CDAMs directly. Three examples are shown in Fig. S14.

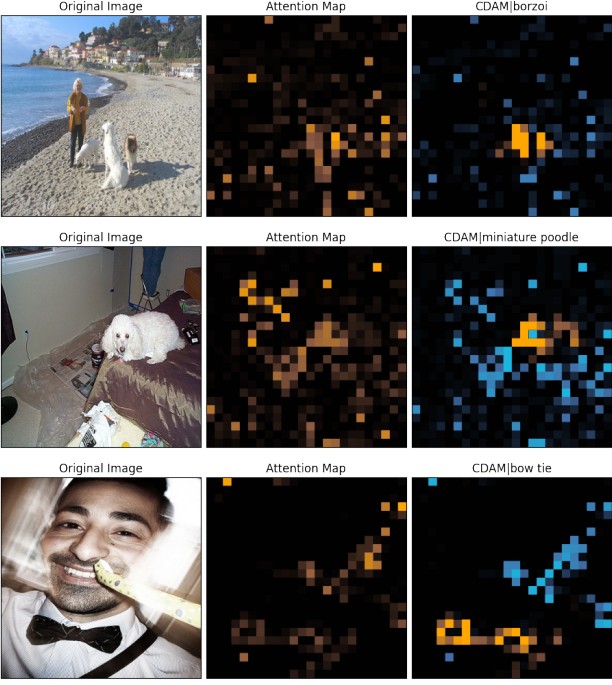

Figure S14: Attention maps and CDAMs from the ImageNet-1K dataset, based on the pre-trained ViT model backbone using SWAG Singh et al. (2022).

In general, ViT models require a massive amount of training data and therefore often trained in self-supervised manner. As shown in DINO (Caron et al., 2021), self-supervised learning can not only improve performance of fine-tuned models, but also improve attention maps. CDAM relies on attention maps, therefore the quality of attention maps strongly influence that of CDAM. Note that the SWAG model has a patch size of $16 \times 16$, which results in lower-resolution attention maps and CDAMs.

### A.5.2 Alternative ViT models for the LIDC

We demonstrate the use of CDAM with ViT models trained using Data-efficient image Transformers (DeIT) (Touvron et al., 2021) and DINOv2 (Oquab et al., 2024). Note that despite the use of the same acronym, DINOv2 is substantially different from DINO (Caron et al., 2021) due to using a different training strategy introduced in iBOT (Zhou et al., 2022) on a much larger scale (Oquab et al., 2024). The original DINO (Caron et al., 2021) was pre-trained on the ImageNet, whereas DINOv2 (Oquab et al., 2024) used a newly curated dataset LVD-142M. Additional tokens called "registers" were introduced to improve attention maps of DINOv2 (Darcet et al., 2024).

The LIDC dataset was split into 5 folds and stratified according to malignancy status. The best-performing models were chosen based on means of Accuracy (ACC) and Mean Squared Error (MSE) over 5 folds, in the validation set. In a parameter sweep, we varied the number of trainable layers (10 - all), dropout rates in the backbone (0.0 - 0.12), batch size (8 - 32) and learning rate scheduler parameters. The learning rate was scheduled with CyclicLR scheduler. Binary Cross Entropy Loss and Huber Loss were used to train classification and regression models, respectively. Weights optimization was performed with Adam optimizer and random rotation was applied as data augmentation. For malignancy classification, the best-performing models based on DeIT and DINOv2 achieved a mean ACC of 0.896 and 0.904, respectively. For biomarker regression, the best-performing models based on DeIT and DINOv2 archived the mean MSE of 0.409 and 0.35, respectively.

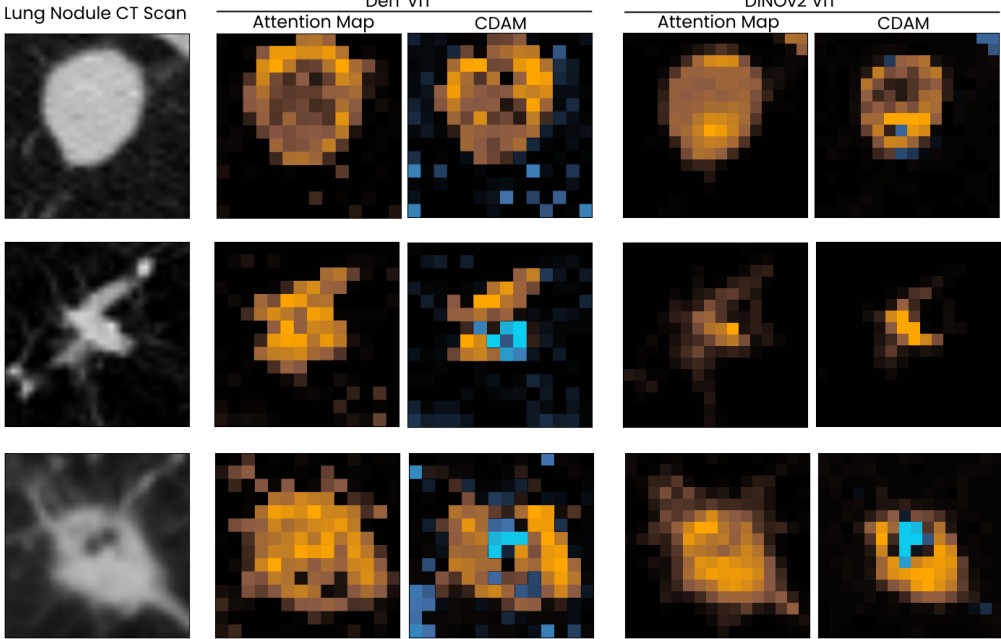

Figure S15: Attention Maps and CDAMs for malignancy models using ViT backbones pre-trained with DeIT and DINOv2.

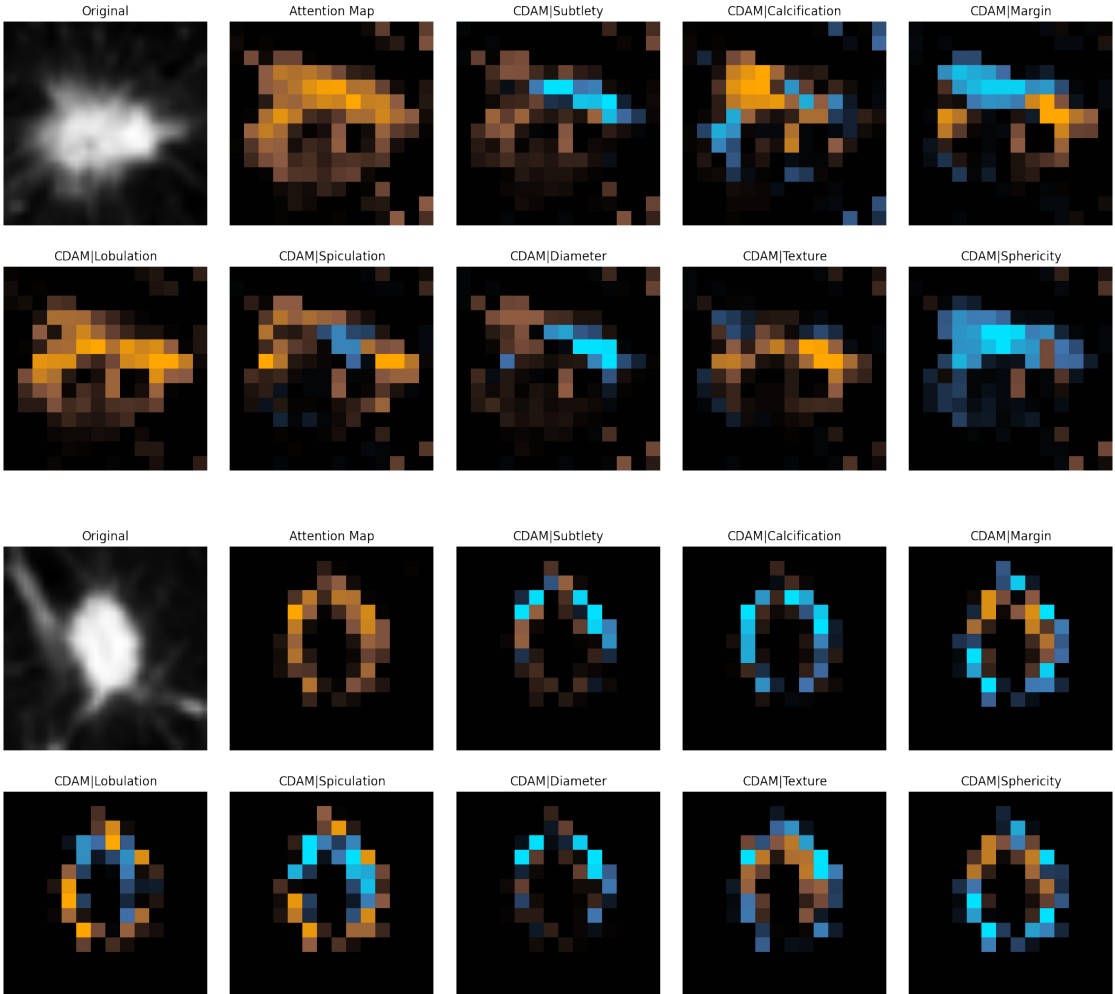

Figure S16: Attention Maps and CDAMs for biomarkers models using ViT backbones pre-trained with DeIT

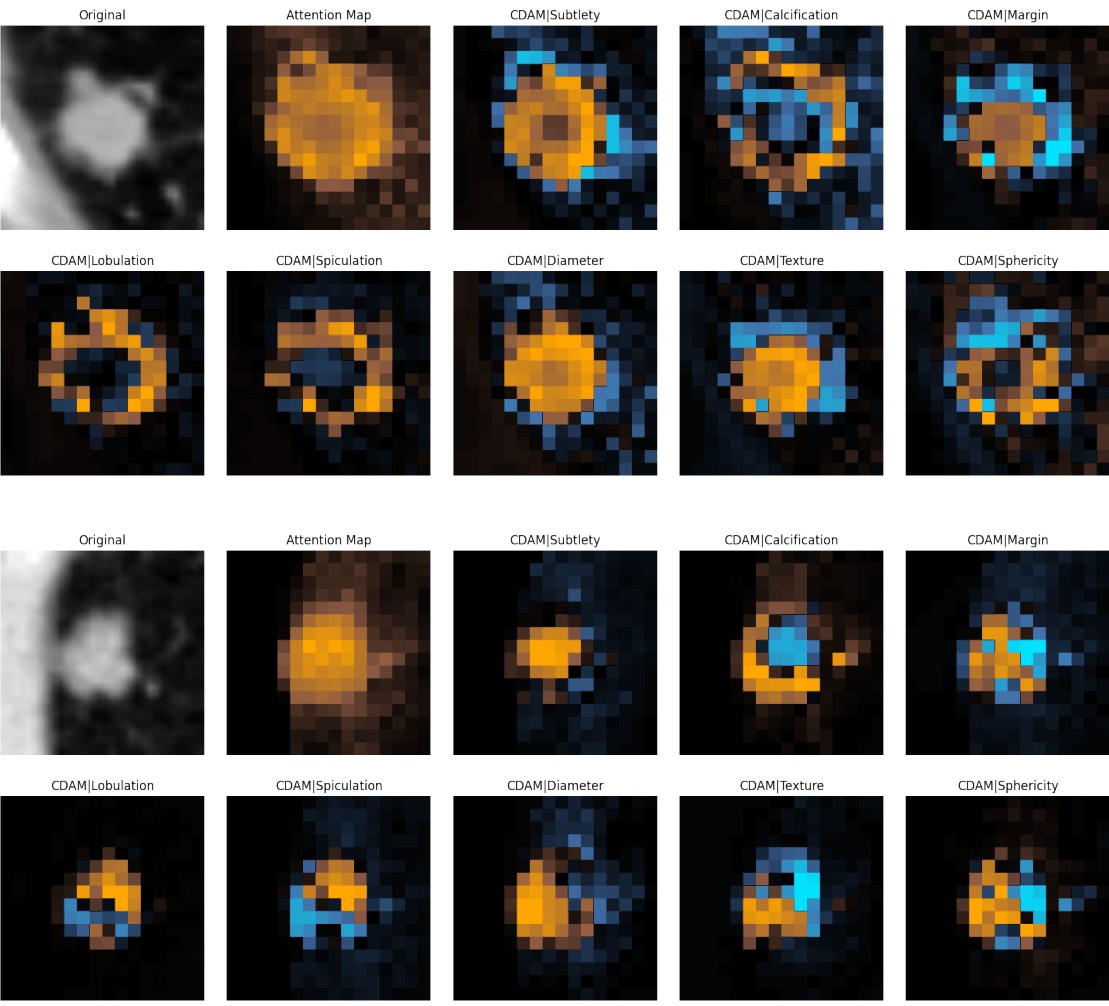

Figure S17: Attention Maps and CDAMs for biomarkers models using ViT backbones pre-trained with DINOv2

