# OpenReview forum: "Class-Discriminative Attention Maps for Vision Transformers"
_TMLR — Accepted by TMLR_

### Review · Reviewer_UH7P · 2024-08-15

**Summary Of Contributions:**

The submission studies interpretability of Vision Transformers by computing importance as the gradients of the final predictor with respects to the tokens in the last attention-layer of the model. Experiments are performed to evaluate correctness, class-sensitivity, and sparsity of the produced saliency maps on subsets of the ImageNet dataset and the LIDC dataset.

**Audience:**

Yes

**Claims And Evidence:**

No

**Requested Changes:**

> Averaging the latent representations of the example images yields a concept embedding, which is analogous to a concept vector in variational autoencoders (VAE) and related generative models (Brocki & Chung, 2019; Kim et al., 2018)

Could the authors expand on this claim and clarify in which way this is analogous to concept activation vectors (CAVs) as in Kim et al.? CAVs are the directions of separating hyperplanes in the representation space of a deep neural network. Here, the concept representation is defined as the centroid of images with a certain concept of interest. In the general case, these two concept representations may be different.

---

> In that regard, our method is related to GradCam, which backpropagates the gradients to the final convolutional feature map in a CNN.

It is true that GradCAM was introduced in the context of CNNs, but it can be readily applied to ViT architectures (see, for example `pytorch-grad-cam/blob/master/tutorials/vision_transformers.md`). Similarly, other class-discriminative notions of importance have been applied to ViTs (see, Covert et al. "Learning to Estimate Shapley Values with Vision Transformers" [2022]). Could the authors expand on their choice not to compare with these methods? Comparing with these methods may help readers place the proposed methods in the context of existing literature.

---

> By design, attention maps are not class-discriminative since they do not take into account any signal coming from a downstream task.

This claim may be too strong in its current formulation. It is true that a self-supervised ViT is independent of the downstream task. But, after finetuning, the weights of the model may be updated according to a loss function that does depend on the downstream task. So, since the attention weights depend on the model parameters, and the model parameters depend on the downstream task, the attention weights do depend on the downstream task.

---

**Eq. (1)**

* $T_i$ is mentioned in the text but it is not defined in the equation.
* Similarly, $Q$ is mentioned in the text but the equation only contains $Q_{CLS}$.
* Appendix A.1 contains several elements that have not been defined at this point in the main text. I would suggest moving the cross-ref.

---

**Appendix A.1**

Could the authors expand on why the discussion focuses on Eq. (4) only (i.e., concept activation), rather than the general case of any linear classifier? Also, $g$ is introduced in the text as a general function, but here, discussion is carried out for the particular case of dot product. What are the implications of using a different similarity function $g$?

Please number all equations before Eq. (10).

---

**Results**

Figures and Tables in support of the main claims of the paper should appear in the main body of the manuscript. Currently, Figures and Tables are placed after the bibliography, as if they were an Appendix. No figures or tables appear in the main body of the submission to corroborate claims.

---

**CDAM for a latent concept**

> by averaging 30 latent representations of images depicting the concept of interest. These 30 images were randomly selected from the corresponding classes in the ImageNet validation set

Could the authors expand on this choice? If concepts are equivalent to classes, then what is the difference between classification experiments and concept experiments? Usually, when talking about semantic concepts, people refer to sub-concepts that make up a class (e.g., class "zebra" and concepts "stripes", "hooves", "tail", ...).

> when targeting the concept zebra, the model deems this part of the trunk to be similar to a zebra. Speculatively, the trunk may be too similar to the zebra’s muzzle or the trunk’s wrinkles may be mistaken for the zebra’s stripes.

I agree with the authors that this claim is highly speculative, and I would suggest removing it because it does not add evidence in support of the main findings of the submission.

---

> Last but not least, while this paper focuses on a ViT model, CDAM is applicable more broadly to all transformer models with self-attention, including large language models (LLMs).

This is a very broad claim that needs refining or removal. The submission places itself on linear classification for bi-directional representations of images, which is a very different task from autoregressive language modeling. If there is evidence in support of the use of CDAM for certain tasks that may be performed with LLMs, then it should be included in the submission.

---

**Minor comments**

* Inconsistent notation for [CLS] token, sometimes [CLS], sometimes [CLS]' (with a prime '). I am not sure I understand the difference.
* Eq. (10), $\nabla_n$ as the $n$-th coordinate of the gradient may be misinterpreted because $\nabla_x$ commonly refers to the gradient with respects to $x$.
* Typo in $T_i = T_i + \mathcal{N}(0, \sigma^2)$, the LHS needs to have a different symbol.
* Please rephrase "We perform quantitative evaluations on two datasets". This paragraph introduces "two datasets" as two subsets of ImageNet, but the the LIDC dataset is also used.
* Please rephrase "For application in medical images [...] of CT scans". The first sentence introduces general "CT scans". And then later, the paragraph talks about the LIDC dataset. Please state from the beginning of the paragraph that the LIDC dataset is used for this experiment.

**Strengths And Weaknesses:**

Strengths:
- the submission is well-motivated
- the choice of evaluation metrics covers known overlooked pitfalls of explanation methods (e.g., "data perturbation")

Weaknesses:
- the organization of the submission could be improved
- certain claims need to be rephrased

I will expand on my points of confusion below and I am looking forward to discussing with the authors.

---

> ### Author Response · Authors · 2024-10-01
>
> According to “Weakness”, we have reorganized the manuscript thoroughly and rephrased our claims to be more specific.
>
> Requested Changes:
> 1. We see similarities between our concept embeddings and CAV (Kim et al., 2018) which use latent representations. Nonetheless, the connection is indirect and we decided to remove that citation.
> 2. According to your suggestion, we have applied GradCAM for ViT (```pytorch-grad-cam/blob/master/tutorials/vision_transformers.md```) in three quantitative evaluations. Tables 1-3 for evaluation metrics now included GradCAM.
>
> Shapley-ViT requires training or fine-tuning the model specifically with masking. While we understand that there are other methods, we are currently using 8 methods in addition to 3 proposed CDAM methods.
>
> 3. We have clarified this by revising this sentence as ```Before fine-tuning, attention maps are not class-discriminative since they do not take into account any signal coming from a downstream task.``` We updated the Method section to explicitly state that after fine-tuning, the weights reflect the downstream task.
>
> 4. **Eq. (1)**
>
> We have amended the text after Eq. (1) to explain the relation of $T_i$, $Q$ and $K$. The cross-ref has been moved to section 4.1.1
>
> 5. **Appendix A.1**
>
> The last paragraph in the Appendix A.1 covers the case of a linear classifier. Generally, we did find the calculations for the dot product to be more instructive and covered them therefore in more detail. Please see a statement after the Eq. (20) in the revised manuscript, the only difference is that the directional derivative now acts on $\sum_k h_k W_{kc}$, i.e. the contribution of $h$ to the activation of class $c$ in the prediction vector (CDAM for a class), instead of $\sum_k h_k l_{c,k}$ (CDAM for a concept).
>
> We don’t expect the outcome to be qualitatively different when choosing a different similarity function $g$, other than a dot product. For example, cosine similarity would introduce an additional normalization factor that does not interfere with the main arguments being made.
>
> All equations are numbered now.
>
> 6. **Results**
>
> According to your comment, we moved the figures and tables to relevant sections in the main text.
>
> 7. **CDAM for a latent concept**
>
> We agree wholeheartedly with your assessment. Concepts are not equivalent to classes. For simplicity and direct comparison, we have utilized the annotated classes in the ImageNet to select 30 images. However, the concepts shared by those images are not identical to the class itself. As you mentioned, we did select 30 images that contain “zebra” – the sub-concepts shared by them are likely "stripes", "hooves", "tail", and others. We have thoroughly revised the Section 4.1.2 CDAM for a latent concept.
>
> Furthermore, to demonstrate sub-concepts, we added additional examples of CDAM, where we manually selected images based on appearances of “stripes” and “tails” (see Fig 6):
>
> > Although we selected a group of images based on annotated classes, which are denoted in n Fig. 5 (right), to illustrate concept-based CDAMs, generally the concepts shared by selected images are not identical to the classes themselves. One could manually provide a set of images with a shared concept that is not originally annotated in the dataset. For example, 10 images that contain \textit{zebras} may share semantic concepts (e.g., stripes and tail), that make up the class \textit{zebra}. We demonstrate this by selecting a set of 10 images that exclusively show \textit{stripes} and another set of 10 images that contain diverse animal \textit{tails}. Shown in Fig. 6, CDAM for \textit{stripes} highlights the body of the zebra, but not the body of the cheetah. On the other hand, CDAM for \textit{tails} tends to focus on the tail of the cheetah. Note that 10 images of diverse animal \textit{tails} include body parts of animals, often rear ends and legs.
>
> We removed that speculative sentence ("...this part of the trunk to be similar to a zebra..."). We also rewrote that paragraph to describe the empirical findings.
>
> 8. The sentence in Discussion about applicability of CDAM to large language models (LLMs) is removed.
>
> Minor comments
> 1. The prime denotes that tokens are after the final transformer block. This is visualized in Figure 2. We further added a sentence in the Method section to clarify this: ```In our notation,  [CLS ] refers to the classification token before and [CLS]’ after the final transformer block.```
> 2. In the revised manuscript, we now denote the n-th coordinate of the gradient  as ∇^{n}. Note that Eq. (10) is now Eq. (14)
> 3. Fixed.
> 4. We have rephrased according to your suggestion.
> 5. We have removed that sentence such that we directly introduce the LIDC dataset now.

---

> > ### Comment · Reviewer_UH7P · 2024-10-09
> > **Thank you for your response**
> >
> > I thank the authors for their careful consideration of all reviewers' comments, which helped improve the quality of presentation and the strength of evidence in the revised version of the paper.
> >
> > ---
> >
> > Minor comments on appendix:
> >
> > 1. Please move "A.2 Quantitative Evaluation" to the top of the next page to avoid an empty blank space at the bottom of page 13.
> > 2. Page 24 is blank, it reads "A.3 Examples of CDAMs from the ImageNet" but there is no content.

---

> > > ### Author Response · Authors · 2024-10-09
> > > **Thank you for helping us improve the paper.**
> > >
> > > We have fixed the Appendix according to your comments. The paper is now updated.

---

### Review · Reviewer_8PW2 · 2024-09-03

**Summary Of Contributions:**

The paper describes a method for post-hoc explaining transformer architectures focusing on providing class-discriminative explanations. The method is an adaption of the gradient-based method GradCam from CNNs to ViTs. Additionally, other straightforward extensions of common  methods for CNNs adapted to ViTs have been described and analyzed. The methods have been analyzed on a single backbone and wrt. to multiple evaluation metrics.

**Audience:**

Yes

**Broader Impact Concerns:**

none present, importance and challenges, e.g., explanations used for attacks could be discussed

**Claims And Evidence:**

No

**Requested Changes:**

- Provide clearer conclusion why CDAM is preferred compared to the other versions (smooth and integrated). Or which method is recommend when. In the current version the take away message from the presented experiments is not very clear.
- Discuss how the proposed method differs from GradCam for ViT, e.g, as used in Chefer et al.,  "Transformer interpretability beyond attention visualization.", CVPR, 2021
- Rearranging the important figures and tables within the main paper.

**Strengths And Weaknesses:**

**Strenghts**

-	Used multiple metrics to analyze the explanations.
-	Addresses transformers networks.
-	Provides class- discriminative explanations

**Weaknesses**

-	The method only explains the linear classification layer and the last transformer block not the full ViT (i.e., has similar limitations as the original GradCam method for CNNs). Unclear how much of the conclusions are mainly about the classification layer than the full transformer architecture. It gives the impression that the explanation is just the weighting of the tokens for the classification, i.e., explaining the downstream application/linear layer and not the full ViT. The main goal of the explanation should be clarified.
-	GradCam and integrated gradient have already been applied to ViTs in other papers (e.g., [a,b]). If the implementation here differs this should be better explained.
-	No clear conclusion from the experiments. From the tables it seems like the smooth version or integrated version perform better. [b] also provides an evaluation framework with ground truth data which could provide further insights.
- Only one architecture has been tested, however the paper claims to be more broadly applicable.
-	Presentation of paper can be improved. Tables and Figures of the main paper are all after the references but not yet in the appendix (making the main paper theoretically quite long as well). It would be easier for the reader to have the relevant data and plots closer to the text describing it. Also the descriptions are partly very lengthy (e.g., conclusion mentions twice the introduction of the  additional variations)


[a] Chefer et al., "Transformer interpretability beyond attention visualization.", CVPR, 2021
[b] Hesse et al., "FunnyBirds: A Synthetic Vision Dataset for a Part-Based Analysis of Explainable AI Methods", ICCV, 2023

---

> ### Author Response · Authors · 2024-09-30
>
> According to your suggestions, we have revised our manuscript by making more specific claims and running additional experiments. Particularly, we now used additional pre-trained ViT models other than DINO (see Appendix A.5)
>
> Weaknesses:
> 1. We agree with your assessment. Accordingly, we have revised the manuscript to clarify that CDAM is about the classification layer and the last transformer block.
>
> In the Introduction
> >Note that the goal of CDAM is to explain the classifier head and the final transformer block
>
> In Method
> >Note that CDAM may not explain the full ViT architecture with multiple transformer blocks. Instead, CDAM focuses on the downstream task and the last transformer block.
>
> 2. Please see our answer to Requested Changes #2
> 3. We have conducted multiple quantitative evaluations to consider different characteristics of explanations. There is no single experiment or metric that could give us a comprehensive and clear answer as to what is the best XAI method. However, using compactness, fidelity, and box sensitivity, CDAM, Integrated CDAM, and Smooth CDAM resulted in substantially better scores than comparison XAI methods. You are correct that sometimes Smooth or Integrated CDAM provides a better score (e.g., In Table 3, box sensitivity metrics for class discrimination are 14.6, 14.8, 13.4 for CDAM, Integrated CDAM, and Smooth CDAM. On the other hand, the fourth-ranked method, Integrated Gradient, has a score of 7.6.)
>
> We understand that it may be better to provide an overall suggestion about how to use our methods – we have added such a guideline in Discussion.
>
> 4. Thanks for your suggestion. We have now used additional ViT models, including the original ViT (Dosovitskiy et al., 2020; for clarity, denoted as “DeIT” based on its training strategy), the SWAG (Singh et al., 2022), and the DINOv2 (Oquab et al., 2024 and Darcet et al. 2024). Note that despite the use of the same acronym, DINOv2 is substantially different from DINO (Caron et al., 2021) due to using a different training strategy introduced in iBOT (Zhou et al., 2022) on a much larger scale. The original DINO (Caron et al., 2021) was pre-trained on the ImageNet, whereas DINOv2 (Oquab et al., 2024) used a newly curated dataset LVD-142M. Please see details and figures in A.5:
>
> >While the main manuscript has used ViT models pre-trained using DINO (Caron et al., 2021), other architectures and pre-trained weights can be used for CDAM. In this section, we show examples of alternative ViT models that are trained in alternative manners. As expected, pre-trained ViT models that produce high-quality attention maps result in high-quality CDAMs. In particular, note that the patch size of DINO (Caron et al., 2021) is 8 × 8 compared to much larger patch sizes used in other backbones.
>
> 5. We have moved main figures and tables into the relevant sections of the main manuscript. We shortened some of the descriptions, including removing a duplicated mention of Integrated and Smooth CDAM in Discussion in the revised manuscript.
>
> Requested Changes:
> 1. We revised the Discussion to provide general suggestions on how to use our three methods – CDAM, Smooth CDAM, and Integrated CDAM. They demonstrate slightly different characteristics, but generally performs superbly compared to the next best option.
> 2. The application of GradCAM as used in Chefer et al. (2021) "Transformer interpretability beyond attention visualization" is different from CDAM because the gradients in GradCAM are calculated with respect to *the outputs of the last attention layer*. I.e., Chefer et al. (2021) writes:
>
> >We note that the last output of a Transformer model (before the classification head), is a tensor v ∈ Rs×d, where the first dimension relates to different input tokens, and only the [CLS] token is fed to the classification head. Thus, performing GradCAM on v will impose a sparse gradients tensor ∇v, with zeros for all tokens, except [CLS].
>
> CDAM, on the other hand, calculates the gradients w.r.t. to *tokens that enter the last attention layer*. Since the [CLS] token that enters the final classification head depends on all the tokens that enter the final attention layer, the gradients w.r.t. to all the input tokens are in general non-zero. We have added this distinction in the Section 2 Related Works:
>
> > CDAM stops backpropagation at the tokens that enter the final attention layer.
>
> > GradCam for ViT backpropages the gradients to the outputs of the final attention layer.
>
> Furthermore, we conducted additional experiments and evaluations using GradCAM for ViT. All of our quantitative evaluation metrics are updated to include GradCAM.
>
> 3. Revised accordingly.
>
> Broader Impact Concerns:
> 1. We added *Broader Impact Statement* after the Discussion. We are open to revise this statement if you have specific concerns.

---

> ### Comment · Reviewer_8PW2 · 2024-10-10
>
> The additonal clarifications and reformulation of some statements as well as the additional experiments have heavily improved the paper and now the content is better aligned with the made claims.
>
> Some remaining minor comments/questions:
> - On page 4, just before 3.1. it mentions only DINO as backbone altough additional architectures are now added.
> - The method are not consistently named, e.g. integrated cdam in text vs. cdam integrated in tables
> - Does integrated cdam still fulfill the axioms of Integrated gradients? A discussion on this would be appreciated.
> - In Figure S15, for the second example the attribution map looks the same as CDAM, maybe mixed up?

---

> > ### Author Response · Authors · 2024-10-11
> > **We have revised the paper according to your comments**
> >
> > The revised paper is uploaded. Thanks for your review.
> >
> > - On page 4, just before 3.1. it mentions only DINO as backbone altough additional architectures are now added.
> >
> > Added
> >
> > - The method are not consistently named, e.g. integrated cdam in text vs. cdam integrated in tables
> >
> > Fixed
> >
> > - Does integrated cdam still fulfill the axioms of Integrated gradients? A discussion on this would be appreciated.
> >
> > The axioms of IG hold with respect to the tokens that enter the final transformer block instead of the input pixels. We've added the following discussion in the Section 3.3 Smooth and Integrated CDAM
> >
> > - In Figure S15, for the second example the attribution map looks the same as CDAM, maybe mixed up?
> >
> > We apologized for the mistake. Figure S15 is fixed. For transparency, we share original figures before re-arranging them for publication: https://drive.google.com/drive/folders/13l-9ssap7GUTq12tZotnq1eVq6HCBPou
> > We plan to share more after the double blind review.

---

### Review · Reviewer_D8Vi · 2024-09-03

**Summary Of Contributions:**

This paper proposes a new attribution method for vision transformers. The proposed method reweights the attention scores by the gradient of the logit of a certain class w.r.t. to token embeddings. Experiments demonstrate that this attribution method shows different heatmaps for different target classes, which means it is highly discriminative. The proposed method is also evaluated on several interpretability metrics.

**Audience:**

Yes

**Claims And Evidence:**

Yes

**Requested Changes:**

1.	It is highly encouraged to put figures and tables in the main text rather than in the appendix. Doing so can greatly enhance the readability of the paper.

2.	Could you explain why CDAM usually produces negative attribution scores on the object other than the target object? For example, in Figure 1, when the target class is “Tench”, the attribution scores on the “Dog” pixels are negative, while the attribution scores on the background pixels are near zero.

3.	Minor. The paragraph below Equation (6), $T_i = T_i + N(0,\sigma^2) \rightarrow T’_i = T_i + N(0,\sigma^2)$. Besides, what is a “null vector”?

**Strengths And Weaknesses:**

Strengths:

1.	The paper is well written and easy to follow.

2.	The proposed method successfully combines the good semantic segmentation-like heatmap provided by attention maps and the discriminative information provided by the gradient of the logit of a certain class.

3.	The visualizations in Figure 1, 3, 4, 5 is clear and show that the proposed attribution map is highly discriminative for different target classes.

Weaknesses:

1.	My main concern is about the generality of experiments. The experiments are mainly conducted on ViT trained with the DINO method. It is not clear whether CDAM still works well on ViTs with other training objectives.

2.	The paper does not compare the proposed method with the Shapley value, which is a popular attribution method due to its good axiomatic properties. Could you also conduct evaluation on the Shapley value method?

---

> ### Author Response · Authors · 2024-09-30
>
> Weakness:
> 1. According to your main concern, we conducted additional experiments with ViT models that are trained with DeIT, SWAG, and DINOv2. Generally, relevant CDAMs are generated using ViT backbones with other training strategies. As expected, the quality of attention maps greatly influence the quality of CDAMs. e.g., we can also generate CDAMs from ViT models that are trained from scratch, but ViT models require a very large training dataset and without that, attention maps are of poor quality. Furthermore, the small patch size of DINO (8 x 8) makes attention maps and CDAMs high resolution, in comparison to DeIT (16 x 16), SWAG (16 x 16), and DINOv2 (14). Details are available in the Appendix A.5 with multiple supplementary figures.
>
> 2. Shapley value for ViT (ViT-Shapley; https://arxiv.org/abs/2206.05282?context=cs.LG) is an excellent explainability method. However, ViT-Shapley requires training or fine-tuning a ViT model with masking. We were unable to make this comparison. Please note that as other reviewer suggested, we decided to add an additional method, GradCAM for ViT (i.e., pytorch-grad-cam/blob/master/tutorials/vision_transformers.md) in our quantitative comparison
>
> Requested Changes:
> 1. Yes, the figures and tables are moved to relevant sections in the main text.
> 2. The ViT-based classifier is trained to discriminate among different objects/classes. We speculate that the objects other than the target object is “evidence against” the target class. In other words, when the target class is “Tench”, the appearance of “Dog” lowers the model output (logit). On the other hand, “the background” likely appears indiscriminately in all samples regardless of classes, and therefore is considered neutral with respect to the target class.
> 3. Fixed. The null vector for the integrated CDAM is a vector of 0’s.

---

> > ### Comment · Reviewer_D8Vi · 2024-10-08
> >
> > I would like to thank the authors for their response. Most of my concerns are addressed.

---

### Author Response · Authors · 2024-10-01

Thank you for your detailed and insightful feedback. We've uploaded a revised manuscript following reviewers' remarks.

---

> ### Comment · Action_Editor_kX4A · 2024-10-03
> **engage with the discussion and submit your recommendations**
>
> Dear reviewers,
>
> Thank you again for your contributions in reviewing this paper.
>
> Now the authors have provided their responses to your comments with a revised version of the paper. Please can you take a look at the response and engage in the discussion with the authors and other reviewers?
>
> Please be reminded that afterwards you will need to submit your recommendations.
>
> Thanks,
>
> AE

---

### Decision · Action_Editor_kX4A · 2024-10-20

**Recommendation:** Accept as is

**Comment:**

In this paper, the authors presented a method for post-hoc explanation of vision Transformers. Specifically, a gradient-based extension called class-discriminative attention maps (CDAM) was proposed, along with a smooth CDAM and an integrated CDAM. Experimental analysis show the effectiveness and the discriminative ability of the proposed method wrt several evaluation metrics. The paper is well-written and easy to follow in general.

Three expert reviewers were invited to comment on the submitted paper, and both strengths and weaknesses were raised i nthe reviewing process. With back-and-forth discussions between the reviewers and authors, along with revisions, all major concerns were well addressed by the authors in the revised version. All three reviewers are satisfied with the revision, and in the end, a consistent positive score (one *Accept* and two *Leaning Accept*) was recommended. Both the AE and reviewers agreed on the interesting approach and contributions made in this paper, which could be of interest to a large potential audience of TMLR. As a result, the AE is please to inform the authors that the paper has been accepted to be published in TMLR.

**Audience:**

Yes, there would be a group of audience in TMLR interested in knowing the findings of this paper. Specifically, the study of the interpretability of Vision Transformers and the proposed class-discriminative attention maps (CDAM) could be of interest to the corresponding researchers.

**Claims And Evidence:**

The claims made in the submission are supported by accurate, convincing and clear evidence.